# Content-Style Identification via Differential Independence

**Subash Timilsina** [1]   **Hoang-Son Nguyen** [1]   **Sagar Shrestha** [1]   **Xiao Fu** [1]

## Abstract

Generative analysis often models multi-domain observations as nonlinear mixtures of domain-invariant content variables and domain-specific style variables. Identifying both factors from unpaired domains enables tasks such as domain transfer and counterfactual data generation. Prior work establishes identifiability under (block-wise) statistical independence between content and style, or via sparse Jacobian assumptions on the nonlinear mixing function, but such conditions can be restrictive in practice. In this work, we introduce content-style differential independence (CSDI), an alternative structural condition requiring that infinitesimal variations in content and style induce orthogonal directions on the data manifold, thereby enabling identifiability even when content and style are dependent and the Jacobian is dense. We operationalize this condition through a blockwise orthogonality constraint on the Jacobian subspaces associated with content and style. To support high-dimensional generative models, we design a stochastic regularizer based on numerical Jacobian approximation, enabling scalable training in settings such as high-resolution image generation. Experiments across multiple datasets corroborate the identifiability analysis and demonstrate practical benefits on counterfactual generation and domain translation.

## 1. Introduction

Multi-domain data is often modeled as mixtures of latent content and style variables (Huang et al., 2018b; Choi et al., 2020; Lee et al., 2020; Wang et al., 2016; Wu et al., 2019; Kong et al., 2022). In this context, content refers to the information shared by data across different domains, while style is domain-specific information. For example, when

[1] School of Electrical Engineering and Computer Science, Oregon State University, Corvallis, Oregon, USA. Correspondence to: Xiao Fu <xiao.fu@oregonstate.edu>.

*Proceedings of the 43rd International Conference on Machine Learning*, Seoul, South Korea. PMLR 306, 2026. Copyright 2026 by the author(s).

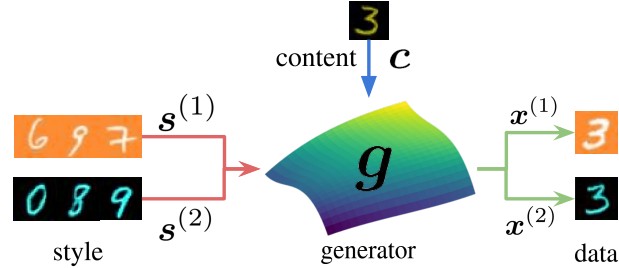

*Figure 1.* Content-style model for multi-domain data—domain 1: digits written in white and orange background; domain 2: digits written in green and black background; content is digit identity, style is background and digit color.

the two domains of images are male and female human faces, shared facial expressions can be considered as content information, while male/female-specific characteristics are controlled by style variables. Identifying latent *content* and *style* variables from multi-domain data has proven useful to many applications, including image translation (Huang et al., 2018b; Lee et al., 2020; Wu et al., 2019; Xie et al., 2023), domain generalization (Kong et al., 2022; Liu et al., 2025), scientific data analysis (Yang et al., 2021), and causal discovery (Sturma et al., 2023; Ahuja et al., 2024).

### 1.1. Existing Approaches and Challenges

Many existing approach for multi-domain content-style identification rely on strong forms of supervision, such as paired or aligned samples across domains (Benaim et al., 2019; Ibrahim et al., 2021; Sørensen et al., 2021; von Kügelgen et al., 2021; Lyu et al., 2022; Karakasis & Sidiropoulos, 2023; Eastwood et al., 2023).

However, such paired data are not always available in practice, particularly in large-scale or naturally occurring datasets, e.g., single-cell data (Yang et al., 2021; Sturma et al., 2023), art stylizations (Zhu et al., 2017), or medical images (Hervella et al., 2019; Yang et al., 2020). This motivates the more challenging *unpaired multi-domain* setting, where samples are only labeled by their domain and no cross-domain correspondences are observed.

Content–style learning from unpaired domains poses a fundamental *identifiability* challenge. In the absence of paired samples, content and style representations may become

arbitrarily entangled or degenerate, yielding distribution-matched solutions with incorrect semantic meaning. To resolve this ambiguity, prior work typically assumes *statistical independence* between content and style, or block-wise independence across domains (Xie et al., 2023; Kong et al., 2022; Shrestha & Fu, 2025). Such assumptions can be restrictive in practice, e.g., in image generation, style of a domain (e.g., illumination) naturally depends on contents (e.g., objects) (von Kügelgen et al., 2021; Schölkopf et al., 2021). More recent approaches impose sparse-influence (or, equivalently, Jacobian sparsity assumptions) on the generative function, requiring style variables to affect fewer features than content variables and their influence supports to be non-overlapping (Yan et al., 2023). As noted in (Nguyen & Fu, 2025), these conditions can be overly stringent for applications such as single-cell genomics data analysis. This raises a fundamental question: *Can content and style be identified from unpaired multi-domain data without assuming statistical independence or Jacobian sparsity?*

### 1.2. Contributions

In this work, we answer this question by using an alternative perspective on content-style disentanglement. Other than enforcing independence at the level of probability distributions, we propose to exploit the notion of *differential independence*. That is, we argue that mutually unaffected infinitesimal variations between content and style are sufficient to encode the notion of content-style disentanglement, which is reminiscent of empirical insights in computer vision (Rolinek et al., 2019; Kumar & Poole, 2020; Peebles et al., 2020; Wei et al., 2021) as well as causal learning frameworks such as independent mechanism analysis (IMA) (Gresele et al., 2021; Reizinger et al., 2022) at block variables levels.

Our detailed contribution is twofold:

*(i) Content-Style Identifiability under Differential Independence.* To represent differential independence between content and style, we impose a *Jacobian-based structural constraint* on the generator, requiring that content-induced and style-induced variations span orthogonal subspaces. This notion of *differential independence* does not require sparse influence of the latent variables on the ambient space or impose statistical independence on the factors, providing an alternative perspective on content-style learning. We show that, under this condition and a distribution-matching loss across domains, both content and style are identifiable up to invertible transformations.

*(ii) Efficient Implementation for High-dimensional Data.* To make our learning loss useful for high-dimensional multi-domain learning (e.g., high-resolution multi-domain image generation and transfer), we develop efficient, scalable implementation of the proposed loss. In particular, we recast

the problem under a multi-domain GAN framework. We propose a spectral random-probing based orthogonality regularization for two subspaces of the learning function's Jacobian, reminiscent of Hutchinson's trace estimation (Hutchinson, 1989). We empirically show that our implementation is scalable, numerically scalable, and effective.

Our identifiability theorem is validated using controlled multi-domain data, as well as high-dimensional multi-domain image transfer and generation tasks.

## 2. Background

### 2.1. Multi-Domain Content-Style Model

We consider a classical multi-domain learning setting, where data is acquired over $N$ domains $\mathcal{X}^{(n)} \subseteq \mathbb{R}^d$, in which $n$ indexes the domain (Xie et al., 2023; Kong et al., 2022; Shrestha & Fu, 2025; Yan et al., 2023). Each data point from domain $n$ is generated from a shared latent variable $\boldsymbol{c}$ (i.e., content) and a domain-specific variable $\boldsymbol{s}^{(n)}$ (i.e., style) through a smooth and invertible generator $\boldsymbol{g}$. For each data point $\boldsymbol{x}^{(n)} \in \mathcal{X}^{(n)} \subseteq \mathbb{R}^d$ coming from a domain $n$,

$$\boldsymbol{x}^{(n)} = \boldsymbol{g}(\boldsymbol{c}, \boldsymbol{s}^{(n)}), \ \boldsymbol{c} \sim p_{\boldsymbol{c}}, \boldsymbol{s}^{(n)} \sim p_{\boldsymbol{s}}^{(n)}, \qquad (1)$$

where the latent representation consists of content variables $\boldsymbol{c} \in \mathcal{C}$ and style variables $\boldsymbol{s}^{(n)} \in \mathcal{S}^{(n)}$ where $\mathcal{C}$ and $\mathcal{S}^{(n)}$ are simply connected open subsets of $\mathbb{R}^{d_C}$ and $\mathbb{R}^{d_S}$. We denote $\mathcal{X} \triangleq \{ \boldsymbol{g}(\boldsymbol{c}, \boldsymbol{s}^{(n)}) : \boldsymbol{c} \in \mathcal{C}, \ \boldsymbol{s}^{(n)} \in \mathcal{S}^{(n)} \} \subseteq \mathbb{R}^d$ and $\mathcal{S} \triangleq \cup_{n=1}^{N} \mathcal{S}^{(n)}$. The generator $\boldsymbol{g}$ is smooth (first-order differentiable) and *bijective* in $(\boldsymbol{c}, \boldsymbol{s}^{(n)})$ on its domain, so that $(\boldsymbol{c}, \boldsymbol{s}^{(n)})$ is uniquely determined by $\boldsymbol{x}^{(n)} = \boldsymbol{g}(\boldsymbol{c}, \boldsymbol{s}^{(n)})$. Under this setting, we have $d \geq d_C + d_S$ (von Kügelgen et al., 2021; Daunhawer et al., 2023).

Fig. 1 presents an illustration of the generative model in (1). There, the domains are handwritten digits with different color, where $\boldsymbol{c}$ represents the digit identity, $\boldsymbol{s}^{(n)}$ the writing style and background color, and $\boldsymbol{g}$ maps the latent codes to the pixel space. This model has been widely adopted in multi-domain representation learning literature, serving for cross-domain counterfactual image generation (Huang et al., 2018b; Lee et al., 2020; Wu et al., 2019), text style transfer (John et al., 2019; Yang et al., 2018; Li et al., 2019), and domain adaptation (Kong et al., 2022; Liu et al., 2025).

### 2.2. Content-Style Identifiability

Under Eq. (1), we hope to identify content $\widehat{\boldsymbol{c}}$ and style $\widehat{\boldsymbol{s}}^{(n)}$ from $\{\boldsymbol{x}^{(n)}\}$. Prior work has studied this identification problem under various settings.

**Aligned Domains.** When content-shared samples $\{\boldsymbol{x}_\ell^{(1)}, \ldots, \boldsymbol{x}_\ell^{(N)}\}$ are aligned (where $\ell$ represents sample index), it was shown that learning criteria that enforces $\boldsymbol{f}(\boldsymbol{x}_\ell^{(i)}) = \boldsymbol{f}(\boldsymbol{x}_\ell^{(j)})$ for all content-shared aligned samples

can provably learn content $\mathbf{c}$ under some mild conditions (von Kügelgen et al., 2021; Lyu et al., 2022; Daunhawer et al., 2023; Karakasis & Sidiropoulos, 2023; Eastwood et al., 2023) . In addition, Lyu et al. (2022); Eastwood et al. (2023) also show identifiability of style $\mathbf{s}^{(n)}$ from such data by exploiting statistical independence between $\mathbf{c}$ and $\mathbf{s}^{(n)}$.

**Unaligned Domains.** In the absence of domain-aligned data, the problem of learning content and style representations becomes more challenging. Most approaches rely on the fact that $\mathbf{c} \sim p_{\mathbf{c}}$ is shared across domains to come up with parameterized distribution matching criteria, e.g., finding $\mathbf{g}$, $\mathbf{c}$ and $\mathbf{s}^{(n)}$ such that

$$\mathbf{x}^{(n)} \overset{(d)}{=\!=\!=} \mathbf{g}(\mathbf{c}, \mathbf{s}^{(n)}), \ \forall n \in [N];$$

see (Xie et al., 2023; Kong et al., 2022; Shrestha & Fu, 2025; Yan et al., 2023). However, distribution matching alone is not sufficient to underpin model identifiability. The works further imposed component-wise statistical independence (Xie et al., 2023; Kong et al., 2022), content-style block independence (Shrestha & Fu, 2025), and Jacobian sparsity of $\mathbf{g}$ (Yan et al., 2023) to assist identification.

**Limitations.** Both content-style statistical independence and Jacobian sparsity have their limitations. In many applications, style naturally depends on content: illumination depends on object geometry, or biological variation depending on cell state, violating statistical independence (von Kügelgen et al., 2021; Schölkopf et al., 2021; Yan et al., 2023). Jacobian sparsity hinges on the assumption that different latent variables emit non-overlapping influences onto subsets of observed features, which may not always hold in applications, e.g., single-cell data analytics (Nguyen & Fu, 2025). These limitations motivate the seek of alternative conditions that enables content-style identification from unaligned domains.

# 3. Proposed Method

To overcome the limitations, we adopt a differential-geometric perspective on the data manifold $\mathcal{X}^{(n)}$. Related geometric structure—explicitly or implicitly exploited in prior work on representation learning and generative modeling—has shown strong empirical promise for learning meaningful and disentangled representations (Rolinek et al., 2019; Kumar & Poole, 2020; Peebles et al., 2020; Gresele et al., 2021; Reizinger et al., 2022; Wei et al., 2021). We formalize this perspective in the setting of unaligned multi-domain learning and provide rigorous theoretical analysis demonstrating how such geometric structure enables identifiability of content and style factors.

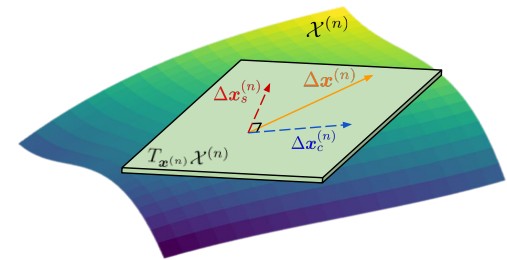

*Figure 2.* An illustration of CSDI assumption: any local data variation $\Delta\mathbf{x}^{(n)} \in T_{\mathbf{x}^{(n)}}\mathcal{X}^{(n)}$ on the data manifold $\mathcal{X}^{(n)}$ is a composition of content variation $\Delta\mathbf{x}_c^{(n)} \in \mathcal{R}(\mathbf{J}_{\mathbf{c}}\mathbf{g}(\mathbf{c}, \mathbf{s}^{(n)}))$ and style variation $\Delta\mathbf{x}_s^{(n)} \in \mathcal{R}(\mathbf{J}_{\mathbf{s}^{(n)}}\mathbf{g}(\mathbf{c}, \mathbf{s}^{(n)}))$, where $\mathcal{R}(\mathbf{J}_{\mathbf{c}}\mathbf{g}(\mathbf{c}, \mathbf{s}^{(n)}))$ and $\mathcal{R}(\mathbf{J}_{\mathbf{s}^{(n)}}\mathbf{g}(\mathbf{c}, \mathbf{s}^{(n)}))$ are mutually orthogonal.

## 3.1. Differentially Independent Content and Style

**Tangent Decomposition.** Intuitively, the data manifold $\mathcal{X}^{(n)}$ is locally formed by mixing content variations and style variations. In particular, for the sample $\mathbf{x}^{(n)} = \mathbf{g}(\mathbf{c}, \mathbf{s}^{(n)})$ in domain $n$, the tangent space at $\mathbf{x}^{(n)}$ characterizes all infinitesimal variations of the data induced by perturbing the latent variables $\mathbf{c}$ and $\mathbf{s}^{(n)}$:

$$T_{\mathbf{x}^{(n)}}\mathcal{X}^{(n)} \triangleq \{\mathbf{J}_{\mathbf{c}}\mathbf{g}(\mathbf{c}, \mathbf{s}^{(n)})\Delta\mathbf{c} + \mathbf{J}_{\mathbf{s}^{(n)}}\mathbf{g}(\mathbf{c}, \mathbf{s}^{(n)})\Delta\mathbf{s}^{(n)} :$$
$$\Delta\mathbf{c} \in \mathcal{C}, \Delta\mathbf{s}^{(n)} \in \mathcal{S}^{(n)}\}$$
$$= \mathcal{R}(\mathbf{J}_{\mathbf{c}}\mathbf{g}(\mathbf{c}, \mathbf{s}^{(n)})) \ \oplus \ \mathcal{R}(\mathbf{J}_{\mathbf{s}^{(n)}}\mathbf{g}(\mathbf{c}, \mathbf{s}^{(n)})),$$

where $\mathcal{R}(\cdot)$ denotes the range space of a matrix, and $\oplus$ denotes the direct sum of two subspaces.

In other words, every local variation $\Delta\mathbf{x}^{(n)}$ on the data manifold $\mathcal{X}^{(n)}$ can be decomposed into content-induced variation $\Delta\mathbf{x}_c^{(n)} \in \mathcal{R}(\mathbf{J}_{\mathbf{c}}\mathbf{g}(\mathbf{c}, \mathbf{s}^{(n)}))$ and style-induced variation $\Delta\mathbf{x}_s^{(n)} \in \mathcal{R}(\mathbf{J}_{\mathbf{s}^{(n)}}\mathbf{g}(\mathbf{c}, \mathbf{s}^{(n)}))$ as $\Delta\mathbf{x}^{(n)} = \Delta\mathbf{x}_c^{(n)} + \Delta\mathbf{x}_s^{(n)}$. Our key assumption imposes an orthogonal relationship between content variations $\Delta\mathbf{x}_c^{(n)}$ and style variations $\Delta\mathbf{x}_s^{(n)}$.

**Assumption 3.1** (Content-Style Differential Independence (CSDI))**.** The content-induced and style-induced tangent subspaces are orthogonal:

$$\mathcal{R}(\mathbf{J}_{\mathbf{c}}\mathbf{g}(\mathbf{c}, \mathbf{s}^{(n)})) \ \perp \ \mathcal{R}(\mathbf{J}_{\mathbf{s}^{(n)}}\mathbf{g}(\mathbf{c}, \mathbf{s}^{(n)})). \qquad (2)$$

Fig. 2 provides an intuitive visualization of CSDI. Assumption 3.1 decouples *local* variations induced by content and style: even if $\mathbf{s}^{(n)}$ may depend on $\mathbf{c}$ statistically, the generator responds to infinitesimal changes in $\mathbf{c}$ and $\mathbf{s}^{(n)}$ through orthogonal directions in data space. The notion of differential independence interprets disentanglement among block factors from a mutually unaffected local-variation perspective, rather than requiring (potentially overly restrictive) statistical independence between $\mathbf{c}$ and $\mathbf{s}^{(n)}$.

*Remark* 3.2. Similar concepts to Assumption 3.1 have repeatedly appeared in representation learning frameworks.

Specifically, Wei et al. (2021) propose explicit regularization that enforces orthogonality among the Jacobian columns of the generative function to disentangle semantic factors. Related disentanglement methods, e.g., object-centric learning (Burgess et al., 2019; Greff et al., 2019; Locatello et al., 2020) and the Hessian penalty (Peebles et al., 2020), also encourage orthogonal Jacobians due to their specific constraints. Orthogonal Jacobian properties are also implicitly promoted in generative models known to induce disentangled representations, including VAEs with diagonal Gaussian priors (Rolinek et al., 2019; Reizinger et al., 2022) and StyleGAN2 with path-length regularization (Karras et al., 2020b). Nonetheless, explicit identifiability guarantees (including content-style identifiability) have not been established in these works.

**CSDI Learning Objective.** Our goal is to identify the model and latent content-style blocks in (1). Using Assumption 3.1, we propose the following learning objective:

$$\text{find invertible } \widehat{\boldsymbol{g}}, \widehat{\boldsymbol{c}}, \{\widehat{\boldsymbol{s}}^{(n)}\}_{i=1}^{N} \tag{3a}$$

$$\text{s.t. } \boldsymbol{x}^{(n)} \overset{\text{(d)}}{=\!=\!=} \widehat{\boldsymbol{g}}(\widehat{\boldsymbol{c}}, \widehat{\boldsymbol{s}}^{(n)}) \tag{3b}$$

$$\boldsymbol{J}_{\widehat{\boldsymbol{c}}}(\widehat{\boldsymbol{g}}(\widehat{\boldsymbol{c}}, \widehat{\boldsymbol{s}}^{(n)})) \perp \boldsymbol{J}_{\widehat{\boldsymbol{s}}^{(n)}}(\widehat{\boldsymbol{g}}(\widehat{\boldsymbol{c}}, \widehat{\boldsymbol{s}}^{(n)})), \ \forall n \in [N]. \tag{3c}$$

Constraint (3b) matches the data distributions of learned generator with the data. Here, distribution matching has to be used as no sample level correspondence (for cross-domain samples sharing the same $\boldsymbol{c} \sim p_{\boldsymbol{c}}$) is known across domains. Constraint (3c) enforces the differential independence by requiring that content- and style-induced variations through decoder $\widehat{\boldsymbol{g}}$ are orthogonal, reflecting Assumption 3.1. Similar distribution matching based criteria were used in (Xie et al., 2023; Kong et al., 2022; Shrestha & Fu, 2025), without the orthogonality constraint in (3c).

Under the learning objective in (3), we now establish identifiability of both ground-truth content $\boldsymbol{c}$ and style $\boldsymbol{s}^{(n)}$.

### 3.2. Identifiability Result

To move forward, we assume the following:

**Assumption 3.3** (Domain Variability). Let $\mathcal{A} \subseteq \mathcal{C} \times \mathcal{S}$ be any measurable set such that $\mathbb{P}_{(\boldsymbol{c},\boldsymbol{s}^{(n)})}[\mathcal{A}] > 0, \forall n \in [N]$ and $\mathcal{A}$ cannot be expressed as $\mathcal{B} \times \mathcal{S}$ for any $\mathcal{B} \subset \mathcal{C}$. There exists a pair of domains $i_{\mathcal{A}}, j_{\mathcal{A}} \in [N]$ such that

$$\mathbb{P}_{(\boldsymbol{c},\boldsymbol{s}^{(i_{\mathcal{A}})})}[\mathcal{A}] \neq \mathbb{P}_{(\boldsymbol{c},\boldsymbol{s}^{(j_{\mathcal{A}})})}[\mathcal{A}]. \tag{4}$$

We remark that for each $\mathcal{A}$, only one pair of domains $i_{\mathcal{A}}, j_{\mathcal{A}}$ is needed to satisfy (4), and $i_{\mathcal{A}}, j_{\mathcal{A}}$ can change with another $\mathcal{A}$. Assumption 3.3 is standard in the content-style identifiability literature (Kong et al., 2022; Xie et al., 2023; Shrestha & Fu, 2025), which requires the styles to have sufficiently

diverse distributions. Intuitively, this assumption implies that two domains $i_{\mathcal{A}}$ and $j_{\mathcal{A}}$ with the same content $\boldsymbol{c}$ are distinguishable due to the domain variation (i.e., styles).

We show the following:

**Theorem 3.4** (Identifiability of Content and Style). *Assume that Eq.* (1) *and Assumption 3.3 hold. Let* $\widehat{\boldsymbol{z}} = (\widehat{\boldsymbol{c}}, \widehat{\boldsymbol{s}}^{(n)})$ *be the latent components learned by solving* (3). *Then, the learned* $\widehat{\boldsymbol{c}}$ *satisfies*

$$\widehat{\boldsymbol{c}} = \boldsymbol{\gamma}(\boldsymbol{c}), \tag{5}$$

*where* $\boldsymbol{\gamma} : \mathcal{C} \to \widehat{\mathcal{C}}$ *is an invertible mapping.*

*If, in addition, Assumption 3.1 holds and the style Jacobian satisfies* $\operatorname{rank}(\boldsymbol{J}_{\boldsymbol{s}^{(n)}}\boldsymbol{g}) = d_S$ *, then for each domain* $n$, *the learned* $\widehat{\boldsymbol{s}}^{(n)}$ *satisfies*

$$\widehat{\boldsymbol{s}}^{(n)} = \boldsymbol{\delta}(\boldsymbol{s}^{(n)}), \qquad \forall n \in [N], \tag{6}$$

*where* $\boldsymbol{\delta} : \mathcal{S} \to \widehat{\mathcal{S}}$ *is an invertible mapping.*

The proof is provided in Appendix B. Note that invertible maps do not lose information of $p_{\boldsymbol{c}}$ and $p_{\boldsymbol{s}}^{(n)}$. Such identification result allows effective content and style-controlled data generation, domain adaptation, and domain transfer, as shown in (Xie et al., 2023; Kong et al., 2022; Shrestha & Fu, 2025; Yan et al., 2023; Timilsina et al., 2024).

Compared to existing content-style identifiability results (Xie et al., 2023; Kong et al., 2022; Shrestha & Fu, 2025; Timilsina et al., 2024; Yan et al., 2023), our conditions provide an alternative with arguably less stringent conditions, without requiring (block or component-wise) statistical independence among latent components as in (Xie et al., 2023; Kong et al., 2022; Shrestha & Fu, 2025; Timilsina et al., 2024) or latent variables' non-overlapping influences onto subsets of features (Yan et al., 2023).

### 3.3. Impact of Inexact Orthogonality

In practice, the orthogonality between content and style Jacobian subspaces in Assumption 3.1 may not hold exactly. In this subsection, we study the impact of violations of Assumption 3.1. To proceed, we consider the following assumption

**Assumption 3.5** (Inexact CSDI). The content-induced and style-induced tangent subspaces are approximately orthogonal. In particular, the smallest principal angle between the two subspaces satisfies

$$\angle\big(\mathcal{R}(\boldsymbol{J}_{\boldsymbol{c}}\boldsymbol{g}), \ \mathcal{R}(\boldsymbol{J}_{\boldsymbol{s}^{(n)}}\boldsymbol{g})\big) \geq \min\{\tfrac{\pi}{2} \pm \xi\} \geq \tfrac{\pi}{2} - \xi,$$

for some $\xi \in [0, \tfrac{\pi}{2}]$.

Using this assumption, we show the following:

**Theorem 3.6** (Robustness of Identifiability). *Assume that Assumption 3.5 holds and that the style Jacobian satisfies* $\operatorname{rank}(\boldsymbol{J}_{\boldsymbol{s}^{(n)}}\boldsymbol{g}) = d_S$. *Then, the following statements hold:*

(a) $\widehat{c} = \gamma(c)$, where $\gamma : \mathcal{C} \to \widehat{\mathcal{C}}$ is an invertible mapping;

(b) $\widehat{s}^{(n)} = \delta(c, s^{(n)})$, where $\delta : \mathcal{C} \times \mathcal{S}^{(n)} \to \widehat{\mathcal{S}}$ is the induced learned-style map, and for every domain $n \in [N]$, the learned $\widehat{s}^{(n)}$ satisfies

$$\left\| J_c \widehat{s}^{(n)} \right\|_2 \leq \tag{7}$$
$$\sin \xi / \sigma_{\min}\left( J_{\widehat{s}} \widehat{g}\left(\widehat{c}, \widehat{s}^{(n)}\right) \right) \| J_c g(c, s^{(n)}) \|_2.$$

The proof is provided in Appendix C.

The theorem shows that inexact orthogonality between $R(J_c g)$ and $\mathcal{R}(J_{s^{(n)}} g)$ does not affect content learning, but will impact style extraction. In particular, when the orthogonality condition is violated with a positive $\xi > 0$, the learned style representation may contain traces of content information. Such content dependence is controlled by three quantities: the non-orthogonality characterized by $\sin \xi$, the magnitude of the ground-truth content Jacobian $\|J_c g\|_2$, and the conditioning of the learned style Jacobian, measured by the smallest non-zero singular value $\sigma_{\min}(J_{\widehat{s}} \widehat{g}(\widehat{c}, \widehat{s}^{(n)}))$.

Note that when $\xi = 0$, the content-induced and style-induced tangent subspaces are exactly orthogonal. In this case, the Theorem 3.6 reduces to Theorem 3.4 as $J_c \widehat{s}^{(n)} = 0$.

# 4. Implementation for High-Dimensional Data

Much of the prior work in this area adopts VAE-based architectures to implement the learning objective; see, e.g., (Kong et al., 2022; Yan et al., 2023). However, VAEs are known to be less suitable for high-dimensional, high-resolution image synthesis (Huang et al., 2018a; Bredell et al., 2023). In contrast, (Xie et al., 2023; Shrestha & Fu, 2025) employ GAN-based backbones, which are better suited for high-resolution image generation and domain transfer. Notably, these methods do not impose constraints on the Jacobian, resulting in relatively straightforward implementations.

**Overall Generative Architecture: CSDI-GAN.** We use a GAN backbone as in (Shrestha & Fu, 2025; Xie et al., 2023). This means that we use two Gaussian vectors $r_C$ and $r_S^{(n)}$ as the "source" of $\widehat{c}$ and $\widehat{s}^{(n)}$, respectively, i.e.,

$$\widehat{c} \triangleq e_C(r_C), \tag{8a}$$
$$\widehat{s}^{(n)} \triangleq e_S^{(n)}(r_S^{(n)}), \forall n \in [N] \tag{8b}$$
$$\widehat{x}^{(n)} \triangleq \widehat{g}(\widehat{c}, \widehat{s}^{(n)}), \forall n \in [N], \tag{8c}$$

where $e_C, \{e_S^{(n)}\}_{n=1}^N$ are trainable invertible content mapping and style mappings, and $\widehat{g}$ is a generating network; see similar structures in (Shrestha & Fu, 2025; Xie et al., 2023). This facilitates easy sampling after model learning.

Unlike (Shrestha & Fu, 2025; Xie et al., 2023) that use independent $r_C$ and $r_S$ (and thus statistically independent $\widehat{c}$

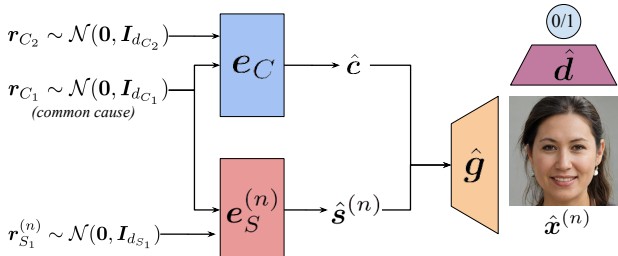

*Figure 3.* CSDI-GAN architecture for content-style learning via differential independence – $e_C, e_S$ are trainable latent mappings, and $\widehat{g}, \widehat{d}$ are generator and discriminator. The face image is randomly generated from https://thispersondoesnotexist.com/.

and $\widehat{s}^{(n)}$), we explicitly model dependent content and style by the following: First, we separately sample two blocks of content, i.e.,

$$r_{C_1} \sim \mathcal{N}(\mathbf{0}, I_{d_{C_1}}), r_{C_2} \sim \mathcal{N}(\mathbf{0}, I_{d_{C_2}}) \tag{9}$$

with $d_{C_1} + d_{C_2} = d_C$, and define $r_C \triangleq (r_{C_1}, r_{C_2})$. Then, for each domain $n$, we sample

$$r_{S_1}^{(n)} \sim \mathcal{N}(\mathbf{0}, I_{d_{S_1}}), \tag{10}$$

and set $r_S^{(n)} \triangleq (r_{C_1}, r_{S_1}^{(n)})$. We then feed these Gaussian vectors to (8) to form $\widehat{x}^{(n)}$—which is then used form distribution matching with $x^{(n)}$ via GANs.

Fig. 3 summarizes our GAN-based architecture. Because $r_{C_1}$ is shared between $r_C$ and $r_S^{(n)}$, $\widehat{c}$ and $\widehat{s}^{(n)}$ can be statistically *dependent* by construction, consistent with Reichenbach's *common cause principle* (Reichenbach, 1956; Schölkopf et al., 2021). Importantly, this dependence is compatible with Assumption 3.1.

**Training Loss.** Based on the generative architecture, we use the following loss to realize our learning criterion in (3):

$$\min_{\widehat{g}, e_C, t_C, \{\widehat{d}^{(n)}, e_S^{(n)}, t_S^{(n)}\}_{n=1}^N} \mathcal{L}_{\text{GAN}}\left(\widehat{g}, e_C, \{e_S^{(n)}\}_{n=1}^N\right) \tag{11}$$

$$+ \lambda_{\text{inv}} \mathcal{L}_{\text{inv}}\left(e_C, t_C, \{e_S^{(n)}, t_S^{(n)}\}_{n=1}^N\}_{n=1}^N\right) + \lambda_{\text{orth}} \mathcal{L}_{\text{orth}}.$$

Here the GAN loss is defined as

$$\mathcal{L}_{\text{GAN}} = \sum_{n=1}^N \mathbb{E}_{x^{(n)}}\left[ \log \widehat{d}^{(n)}(x^{(n)}) \right]$$
$$+ \sum_{n=1}^N \mathbb{E}_{r_C, r_S^{(n)}}\left[ \log\left(1 - \widehat{d}^{(n)}(\widehat{x}^{(n)})\right) \right], \tag{12}$$

in which $\widehat{x}^{(n)} = \widehat{g}(e_C(r_C), e_S^{(n)}(r_S^{(n)}))$ and $\widehat{d}^{(n)}$ is a discriminator for domain $n$. This term is used to realize (3b). The invertibility loss is defined as

$$\mathcal{L}_{\text{inv}} = \mathbb{E}\left[ \| t_C(e_C(r_C)) - r_C \|_2^2 \right]$$
$$+ \sum_{n=1}^N \mathbb{E}\left[ \| t_S^{(n)}(e_S^{(n)}(r_S^{(n)})) - r_S^{(n)} \|_2^2 \right], \tag{13}$$

with $\boldsymbol{t}_C$ and $\boldsymbol{t}_S^{(n)}$ being invertibility-promoting decoders (inverses) for the corresponding encoders $\boldsymbol{e}_C$ and $\boldsymbol{e}_S^{(n)}$. The parameter $\lambda_{\mathrm{inv}} > 0$ in (11) controls the weight of the invertibility regularization. This constraint implicitly encourages the learned generator $\widehat{\boldsymbol{g}}$ to be invertible. In particular, combining this criterion (13) with the arguments in (Shrestha & Fu, 2025; Zimmermann et al., 2021) shows that under the latent dimensionality knowledge, GAN-based distribution matching yields an bijective mapping between the latent manifold and the data manifold; see Appendix A for details.

The last term $\mathcal{L}_{\mathrm{orth}}$ is used to approximate orthogonality in (3c) and $\lambda_{\mathrm{orth}} > 0$ balances the regularization's importance. The orthogonality between Jacobian subspaces is quite nontrivial to enforce. The reason is twofold. First, for high-dimensional data, evaluating Jacobian at even a single sample would require a large number of forward- and back-propagation passes, which is not affordable—and extending this to all samples is prohibitive. Second, measuring subspace angles needs proper Jacobian normalization to prevent numerical issues (otherwise close-to-zero Jacobian columns could create small inner products and thus "false orthogonality"). However, normalization at scale is also a challenging issue. To proceed, we propose using the following regularization:

$$\mathcal{L}_{\mathrm{orth}} \triangleq \sum_{n=1}^{N} \mathbb{E}_{\widehat{\boldsymbol{c}}, \widehat{\boldsymbol{s}}^{(n)}} \left[ \frac{\left\| \boldsymbol{J}_{\widehat{\boldsymbol{s}}^{(n)}}^{\top} \boldsymbol{J}_{\widehat{\boldsymbol{c}}} \right\|_{\mathrm{F}}^2}{\left\| \boldsymbol{J}_{\widehat{\boldsymbol{c}}} \right\|_{\mathrm{F}}^2 \left\| \boldsymbol{J}_{\widehat{\boldsymbol{s}}^{(n)}} \right\|_{\mathrm{F}}^2 + \epsilon} \right], \quad (14)$$

where $\boldsymbol{J}_{\widehat{\boldsymbol{c}}} \triangleq \boldsymbol{J}_{\widehat{\boldsymbol{c}}} \widehat{\boldsymbol{g}}(\widehat{\boldsymbol{c}}, \boldsymbol{s}^{(n)})$ and $\boldsymbol{J}_{\widehat{\boldsymbol{s}}^{(n)}} \triangleq \boldsymbol{J}_{\widehat{\boldsymbol{s}}^{(n)}} \widehat{\boldsymbol{g}}(\widehat{\boldsymbol{c}}, \widehat{\boldsymbol{s}}^{(n)})$. A small constant $\epsilon > 0$ is introduced in the denominator to ensure numerical stability when $\|\boldsymbol{J}_{\widehat{\boldsymbol{c}}}\|_{\mathrm{F}}, \|\boldsymbol{J}_{\widehat{\boldsymbol{s}}^{(n)}}\|_{\mathrm{F}}$ are small. Minimizing $\mathcal{L}_{\mathrm{orth}}$ enforces content-style Jacobian orthogonality by encouraging $\left\| \boldsymbol{J}_{\widehat{\boldsymbol{s}}^{(n)}}^{\top} \boldsymbol{J}_{\widehat{\boldsymbol{c}}} \right\|_{\mathrm{F}}$ to be close to 0, while avoiding the trivial solution $\boldsymbol{J}_{\widehat{\boldsymbol{c}}} \approx \boldsymbol{0}, \boldsymbol{J}_{\widehat{\boldsymbol{s}}^{(n)}} \approx \boldsymbol{0}$ or numerical stability issues (see Appendix D.1.1 for details).

To efficiently compute $\mathcal{L}_{\mathrm{orth}}$, we use Hutchinson's trace estimator (Hutchinson, 1989) to obtain unbiased estimates for each term $\boldsymbol{J}_{\widehat{\boldsymbol{s}}^{(n)}}^{\top} \boldsymbol{J}_{\widehat{\boldsymbol{c}}}, \|\boldsymbol{J}_{\widehat{\boldsymbol{c}}}\|_{\mathrm{F}}^2, \|\boldsymbol{J}_{\widehat{\boldsymbol{s}}^{(n)}}\|_{\mathrm{F}}^2$ in (14); details are provided in Appendix D.1.2. This yields the following approximation for each summand of (14):

$$\frac{\|\boldsymbol{J}_{\widehat{\boldsymbol{s}}^{(n)}}^{\top} \boldsymbol{J}_{\widehat{\boldsymbol{c}}}\|_{\mathrm{F}}^2}{\|\boldsymbol{J}_{\widehat{\boldsymbol{c}}}\|_{\mathrm{F}}^2 \|\boldsymbol{J}_{\widehat{\boldsymbol{s}}^{(n)}}\|_{\mathrm{F}}^2 + \epsilon} \approx \frac{\|\mathbb{E}_{\boldsymbol{v}} [\boldsymbol{J}_{\widehat{\boldsymbol{s}}^{(n)}}^{\top} \boldsymbol{v}(\boldsymbol{J}_{\widehat{\boldsymbol{c}}}^{\top} \boldsymbol{v})^{\top}]\|_{\mathrm{F}}^2}{\mathbb{E}_{\boldsymbol{v}} [\|\boldsymbol{J}_{\widehat{\boldsymbol{c}}}^{\top} \boldsymbol{v}\|_2^2] \mathbb{E}_{\boldsymbol{v}} [\|\boldsymbol{J}_{\boldsymbol{s}^{(n)}}^{\top} \boldsymbol{v}\|_2^2] + \epsilon},$$
$$(15)$$

where $\boldsymbol{v} \in \mathbb{R}^d$ is a random vector with $\mathbb{E}[\boldsymbol{v}\boldsymbol{v}^{\top}] = \boldsymbol{I}_d$. This approximation allows us to use vector-Jacobian products (VJP) to efficiently calculate (15) without storing the whole Jacobians in memory:

$$\boldsymbol{J}_{\widehat{\boldsymbol{c}}}^{\top} \boldsymbol{v} = \nabla_{\widehat{\boldsymbol{c}}} \langle \widehat{\boldsymbol{g}}(\widehat{\boldsymbol{c}}, \widehat{\boldsymbol{s}}^{(n)}), \boldsymbol{v} \rangle,$$
$$\boldsymbol{J}_{\widehat{\boldsymbol{s}}^{(n)}}^{\top} \boldsymbol{v} = \nabla_{\widehat{\boldsymbol{s}}^{(n)}} \langle \widehat{\boldsymbol{g}}(\widehat{\boldsymbol{c}}, \widehat{\boldsymbol{s}}^{(n)}), \boldsymbol{v} \rangle. \quad (16)$$

*Remark* 4.1. Directly evaluating (14) is computationally impractical for high-dimensional data, because explicitly forming $\boldsymbol{J}_{\widehat{\boldsymbol{c}}}$ and $\boldsymbol{J}_{\widehat{\boldsymbol{s}}^{(n)}}$ would require $\mathcal{O}(B\,d(d_C + d_S))$ memory for batch size $B$ and $\mathcal{O}(B\,d)$ backward passes (one per output coordinate). We therefore adopt the noise-probing VJP estimator (16), which replaces the $\mathcal{O}(d)$ factor by $\mathcal{O}(K)$ with $K \ll d$ being number of sampled probes $\boldsymbol{v}$. This avoids storing the full $d \times d_C$ and $d \times d_S$ Jacobians and reducing the Jacobian-related memory by roughly a factor of $(d_C + d_S)$ (see Appendix D.1.3 for more details).

*Remark* 4.2. Hutchinson-style noise probing is widely used to efficiently estimate Jacobian-based regularizers without explicitly forming Jacobians (Karras et al., 2020b; Peebles et al., 2020; Wei et al., 2021). However, implementing (3c) presents unique challenges that the existing techniques cannot resolve. Path length regularization from Karras et al. (2020b) relies on VJPs, but it only controls the Jacobian of a *single* latent variable and does not capture the *cross*-interaction between content and style required for (3c). (Peebles et al., 2020; Wei et al., 2021) use finite difference, which requires multiple forward passes per sampled directions and becomes expensive when applied across many blocks $\boldsymbol{J}_{\widehat{\boldsymbol{c}}}, \boldsymbol{J}_{\widehat{\boldsymbol{s}}^{(1)}}, ..., \boldsymbol{J}_{\widehat{\boldsymbol{s}}^{(N)}}$ as needed in our context. Furthermore, this approximation introduces extensive hyperparameter search, due to using $N + 1$ noise probes and tuning $N + 1$ step sizes for each Jacobians.

## 5. Related Works

**Identifiable Representation Learning with Jacobian Regularizers.** Regularization on Jacobian has been an important means to establish identifiability in representation learning; see, e.g., (Locatello et al., 2019; Zheng & Zhang, 2023). Nguyen & Fu (2025) proposes a Jacobian volume maximization approach to recover latent variables, which achieves identifiability under reasonable conditions. Another line of works relies on Jacobian sparsity for identifiability, either via structured sparse influence (Zheng & Zhang, 2023; Moran et al., 2022), compositional/object-centric assumption (Lachapelle et al., 2023; Brady et al., 2023; 2025). These methods are later generalized by (Matthes et al., 2026). The IMA work (Gresele et al., 2021) uses Jacobian orthogonality to identify latent components, yet understanding to its identifiability remains incomplete; see (Buchholz et al., 2022). These models care about elementwise latent component identification (other than content-style block identification) and often do not consider multi-domain data.

**Orthogonal Jacobian in Generative Models.** Deep generative models have explored orthogonal Jacobian structures from various angles. Rolinek et al. (2019) shows that VAEs with diagonal Gaussian prior promote a decoder with orthogonal Jacobian, empirically leading to better identification of disentangled latent components. StyleGAN2 (Karras et al., 2020b) training employs the so-called path length

regularization, which promotes orthogonality of the generator's Jacobian. The design intention was for latent space smoothness and image quality, but it leads to properties such as latent component semantic disentanglement. Wei et al. (2021) confirmed this observation by explicitly imposes a orthogonal Jacobian constraints on GANs. Other disentanglement methods, e.g., object-centric learning (Burgess et al., 2019; Locatello et al., 2020) and Hessian Penalty (Peebles et al., 2020), also implicitly encourage Jacobian orthogonality.

**Identifiable Content-Style Learning from Unpaired Data.**
Content-style identification from unpaired data is widely used for image translation (Shrestha & Fu, 2025; Xie et al., 2023), counterfactual generation (Yan et al., 2023), domain adaptation (Kong et al., 2022; Timilsina et al., 2024), and causal representation learning (Sturma et al., 2023). In terms of identifiability, Xie et al. (2023); Kong et al. (2022) assume that all latent components of $z^{(n)} = (c, s^{(n)})$ are statistically independent given domain label $n$, and there should be at least $2d_s + 1$ domains. Shrestha & Fu (2025) uses a block independence condition $p(c, s^{(1)}, ..., s^{(n)}) = p(c)p(s^{(1)})...p(s^{(n)})$ and can work with only two domains. However, statistical independence between content-style might be restrictive in many applications where style naturally depends on content (von Kügelgen et al., 2021; Schölkopf et al., 2021). The work (Yan et al., 2023) proposes to learn dependent content and style by exploiting Jacobian sparsity. However, such sparse Jacobian structure may not always hold in applications where latent variables densely influence the data; e.g., in single-cell data (Nguyen & Fu, 2025). Our method provides alternative identifiability conditions based on the notion of differential independence, expanding scenarios where content and style can be provably identified.

# 6. Numerical Results

**Two-domain MNIST Dataset.** We construct a controlled two-domain variant of MNIST to test identification of dependent content-style. The digit identity (content) is shared across domains, while the style is domain-specific: *digit color in Domain 1* and *background color in Domain 2*, sampled from label-dependent mixtures to induce tunable dependence; see Fig. 9 and other details of the data generation process in Appendix E.2.

We implement the content-style generative model (8) using DCGAN (Radford et al., 2016), with $\mathcal{L}_{\text{GAN}} + \mathcal{L}_{\text{inv}}$ as training loss. See Appendix E.2 for hyperparameter settings. We compare our proposed content-style dependence model (i.e., (8) using dependent $\widehat{c}, \widehat{s}^{(n)}$) against an existing content-style block independence GAN model (B.I. GAN) in (Shrestha & Fu, 2025). To illustrate the effect of differential independence regularizer, we compare our CSDI-GAN models when trained with/without $\mathcal{L}_{\text{orth}}$.

*Varying Content*

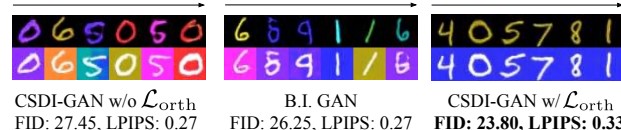

| CSDI-GAN w/o $\mathcal{L}_{\text{orth}}$ | B.I. GAN | CSDI-GAN w/ $\mathcal{L}_{\text{orth}}$ |
|---|---|---|
| FID: 27.45, LPIPS: 0.27 | FID: 26.25, LPIPS: 0.27 | **FID: 23.80, LPIPS: 0.33** |

*Figure 4.* Generation task on two-domain MNIST data with same $\widehat{s}^{(n)}$ and varying $\widehat{c}$: CSDI-GAN w/o $\mathcal{L}_{\text{orth}}$, B.I. GAN, CSDI-GAN w/ $\mathcal{L}_{\text{orth}}$ (Proposed).

***Counterfactual Generation.*** Fig. 4 reports the results for counterfactual generation task, using the learned generator $\widehat{g}$. For Fig. 4, we fix the styles $\widehat{s}^{(1)}, \widehat{s}^{(2)}$ and randomly sample multiple (six) contents $\widehat{c} = e_C(r_C), r_C \sim \mathcal{N}(0, I_{d_C})$ to create multiple (six) images $\widehat{x}^{(n)} = \widehat{g}(\widehat{c}, \widehat{s}^{(n)})$. Fig. 4 shows that CSDI-GAN can identify the ground-truth $c, s^{(n)}$, which are used to accurately generate images of the *fixed style* with *different contents* in a consistent way. In contrast, the baselines learned an entangled $\widehat{c}, \widehat{s}^{(n)}$, which makes them fail to consistently generate images with same style.

We also measure quality of generated images using FID ($\downarrow$) score (Heusel et al., 2017), and use LPIPS ($\uparrow$) distance (Zhang et al., 2018) between pairs of images from the same $\widehat{c}$ with different $\widehat{s}^{(n)}$ to measure style diversity; see Appendix E.2 for details. As shown in Fig. 4, our method surpasses the baselines in terms of both metrics.

***Domain Translation.*** Given trained generator $\widehat{g}$ and style mapping $e_S^{(n)}$, translation is done via two steps. First, we solve for $(\widehat{c}, \widehat{s}^{(i)}) \triangleq \arg\min_{\tilde{c}, \tilde{s}^{(i)}} \text{div}(\widehat{g}(\tilde{c}, \tilde{s}^{(i)}), x^{(i)})$, where $\text{div}(\cdot, \cdot)$ is a divergence measure, to obtain the content-style representation of an image $x^{(i)}$ (akin to GAN inversion (Xia et al., 2023)). Similarly, we extract the style $\widehat{s}^{(t)}$ from a target domain image. Second, we combine the extracted content $\widehat{c}$ from the source image with the target style $\widehat{s}^{(t)}$ to create the translated $\widehat{x}^{(t)} = \widehat{g}(\widehat{c}, \widehat{s}^{(t)})$. More details on translation procedure are provided in Appendix E.2.

Fig. 5 shows a qualitative result for translation. Each row indicates a content that we want to preserve in translation to a target domain with a new style. The baselines fail to identify ground-truth $c, s^{(n)}$ and thus output entangled, low-quality translated images. Our method clearly learns the content and style, outputting intended translations. For further experiments, see Appendix. E.2

**AFHQ and CelebA-HQ Datasets.** We further verify our identifiability result on real-world datasets: animal faces (AFHQ) with 3 domains (dog/cat/wild animals) (Choi et al., 2020), and human faces (CelebA-HQ) with 2 domains (male/female) (Karras et al., 2018). In addition to conventional GAN models StyleGAN2-ADA (Karras et al., 2020a) and Transitional-cGAN (Shahbazi et al., 2022), we compare our CSDI-GAN with baselines specifically designed for

*Table 1.* Comparison of FID (↓) and LPIPS (↑) for generation (left) and translation (right) tasks on AFHQ and CelebA-HQ.

*(a)* Generation task.

| Method | AFHQ | | CelebA-HQ | |
|---|---|---|---|---|
| | FID (↓) | LPIPS (↑) | FID (↓) | LPIPS (↑) |
| Transitional-cGAN | 6.7 | – | 6.4 | – |
| StyleGAN2-ADA | 6.5 | – | 5.0 | – |
| I-StyleGAN (Tr) (Xie et al., 2023) | 5.6 | 0.3436 | 4.8 | 0.2799 |
| B.I. GAN (Shrestha & Fu, 2025) | 5.2 | 0.3995 | 4.6 | 0.2628 |
| CSDI-GAN (w/o $\mathcal{L}_{\text{orth}}$) | 5.3 | 0.4079 | 6.0 | 0.2467 |
| **CSDI-GAN (Proposed)** | **4.4** | **0.4452** | **4.3** | **0.3392** |

*(b)* Translation task.

| Method | AFHQ | | CelebA-HQ | |
|---|---|---|---|---|
| | FID (↓) | LPIPS (↑) | FID (↓) | LPIPS (↑) |
| StarGANv2 (Choi et al., 2020) | 15.0 | 0.3578 | 14.3 | **0.3148** |
| I-StyleGAN (Xie et al., 2023) | 17.6 | 0.3701 | 19.7 | 0.2003 |
| B.I. GAN (Shrestha & Fu, 2025) | 10.5 | 0.4107 | 24.6 | 0.2828 |
| **CSDI-GAN (Proposed)** | **7.1** | **0.4392** | **12.9** | 0.3105 |

Reference Style

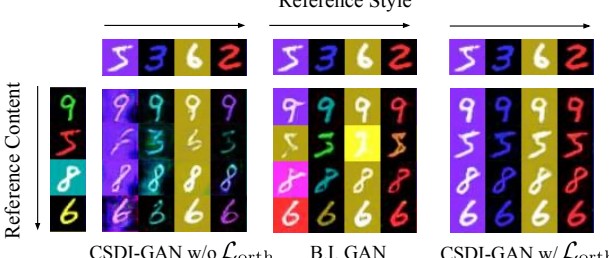

CSDI-GAN w/o $\mathcal{L}_{\text{orth}}$    B.I. GAN    CSDI-GAN w/ $\mathcal{L}_{\text{orth}}$

*Figure 5.* Translation task on two-domain MNIST data: CSDI-GAN w/o $\mathcal{L}_{\text{orth}}$, B.I. GAN, CSDI-GAN w/ $\mathcal{L}_{\text{orth}}$ (Proposed).

content-style learning: I-StyleGAN (Xie et al., 2023), Block Independence GAN (B.I. GAN) (Shrestha & Fu, 2025).

We use StyleGAN2-ADA as the base architecture and train from scratch using the corresponding loss functions from CSDI-GAN, I-StyleGAN (Xie et al., 2023), and B.I. GAN (Shrestha & Fu, 2025). To illustrate the effect of enforcing Assumption 3.1, we also test our CSDI-GAN without using the regularizer $\mathcal{L}_{\text{orth}}$. Further experiment details, hyperparameter settings, and additional qualitative results can be found in Appendix E.3.

***Counterfactual Generation.*** Table 1a shows the image quality (measured by FID) and the style diversity[1] (measured by LPIPS, similar to MNIST experiments) of all considered methods. We can see that the proposed CSDI-GAN outperforms the baselines in both metrics, including the GAN architectures designed for content-style modeling.

Fig. 6 presents several failure cases of the baselines on AFHQ. In this dataset, the content is animal's pose, whereas the style is a breed of dog/cat/wild animal. We observe that B.I. GAN (Shrestha & Fu, 2025) and I-StyleGAN (Xie et al., 2023) fail to keep the style (animal breed) consistent when changing the content (pose); for example, B.I. GAN generates a lion, a wolf, and a leopard, which are three different breeds of animals. In contrast, our CSDI-GAN consistently keep the style information disentangled from content variations (i.e., generating the same cat and the same fox). We

---

[1]As StyleGAN2-ADA and Transitional-cGAN do not learn a content-style model, their style diversity (LPIPS) are not evaluated.

Different sampled content, with fixed style

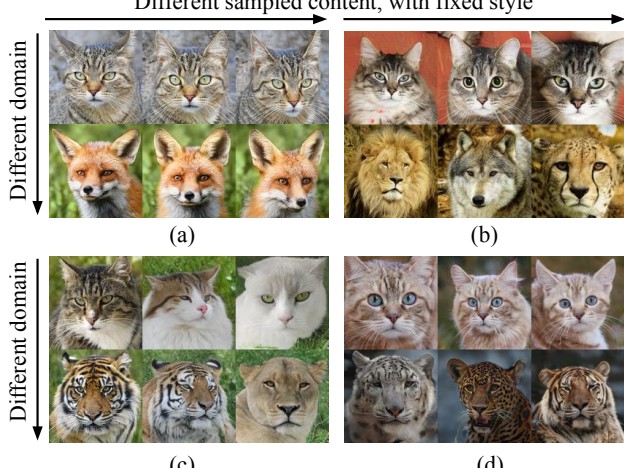

*Figure 6.* Image generation task on AFHQ (a) CSDI-GAN (Ours) (b) B.I. GAN (Shrestha & Fu, 2025) (c) I-StyleGAN (Xie et al., 2023) (d) CSDI-GAN w/o $\mathcal{L}_{\text{orth}}$.

also see the importance of enforcing Assumption 3.1 in content-style learning: CSDI-GAN without $\mathcal{L}_{\text{orth}}$ also fails to preserve the style information and generates two different wild animals (a tiger and a leopard).

***Domain Translation.*** We follow the same domain translation procedure described in our MNIST experiment. Table 1b shows the image quality (FID) and the style diversity (LPIPS, measured in a similar way to MNIST experiments) of the baselines. We also add StarGANv2, which is a *dedicated system* for multi-domain translation without learning content-style representation (Choi et al., 2020). Our proposal CSDI-GAN mostly outperforms the baselines in both metrics, except for the style diversity in CelebA-HQ dataset where CSDI-GAN is on par with StarGANv2.

Fig. 7 shows several resulting translations of the considered methods. Each row presents the result of translating between a source domain (shown by *Content* column) and a target domain (shown by *Style* column). While CSDI-GAN successfully preserves the content and style during translation, the baselines fail in some cases. For example, B.I. GAN produced a wolf instead of a tiger in `cat2wild`, StarGANv2 did not keep the dog breed content in `wild2dog`,

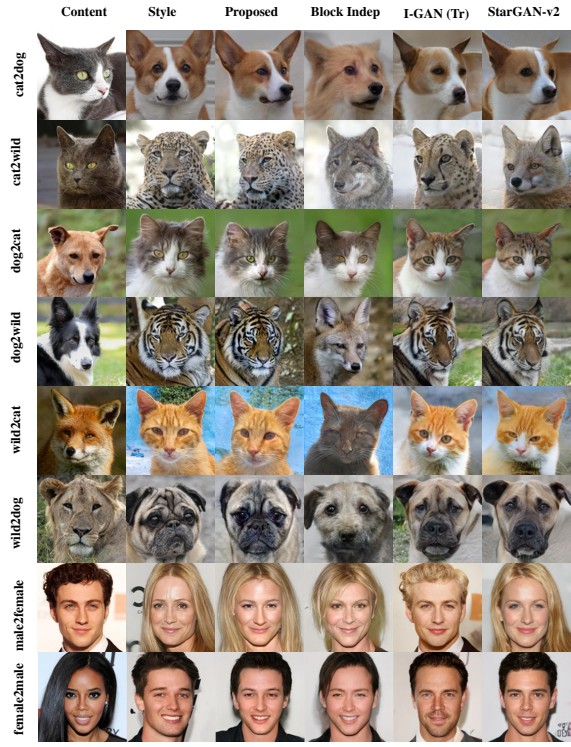

*Figure 7.* Comparing of translation results of CSDI-GAN (Ours), B.I. GAN (Shrestha & Fu, 2025), I-StyleGAN (Xie et al., 2023), and StarGANv2 (Choi et al., 2020).

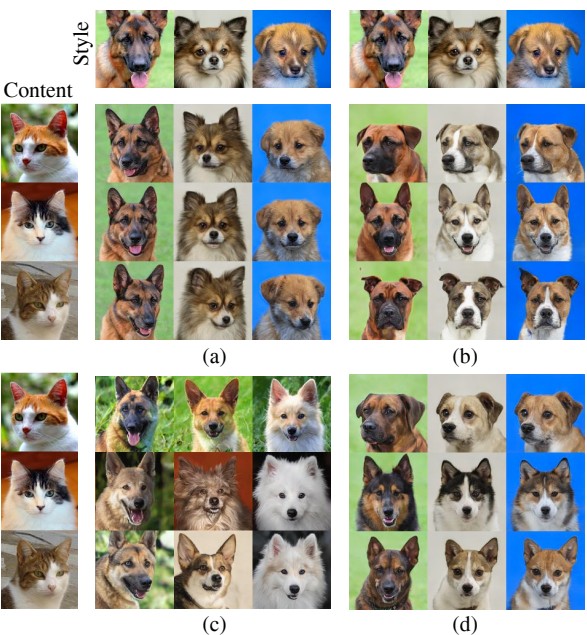

*Figure 8.* cat2dog image translation results in AFHQ dataset. (a) Our CSDI-GAN w/ $\mathcal{L}_{\text{orth}}$ (b) B.I. GAN (Shrestha & Fu, 2025) (c) I-StyleGAN (Xie et al., 2023) (d) CSDI-GAN w/o $\mathcal{L}_{\text{orth}}$.

and I-StyleGAN failed in male2female.

Fig. 8 shows additional qualitative results for cat2dog translation on AFHQ. Each row corresponds to the content to be preserved (i.e., pose), and each column corresponds to a target style (i.e., dog breed). CSDI-GAN with $\mathcal{L}_{\text{orth}}$ successfully preserves the specified pose while translating the image into the dog domain with the desired style. In contrast, B.I. GAN (Shrestha & Fu, 2025) and I-StyleGAN (Xie et al., 2023) exhibit different failure modes. B.I. GAN sometimes fails to preserve the dog breed specified by the style input. I-StyleGAN exhibits similar style failures and also shows a spurious correlation: it tends to preserve the cat-ear shape during translation, producing dog images with similar ear structures. Although B.I. GAN generally preserves pose, I-StyleGAN fails to do so in some cases. Finally, removing $\mathcal{L}_{\text{orth}}$ from CSDI-GAN preserves content reasonably well but often fails to retain the target dog identity. These results confirm that the content-style differential independence regularizer $\mathcal{L}_{\text{orth}}$ plays a critical role in extracting style information.

For reproducibility, the implementation of CSDI-GAN is publicly available at: https://github.com/subashtimilsina/CSDI.

# 7. Conclusion

We revisit the problem of content-style identifiability from unpaired multi-domain data, which is the foundation behind various applications such as counterfactual generation and domain translation. Departing from previous modeling principles, we show that provable identification of both content and style can be achieved via a learning objective based on differential independence between $c$ and $s^{(n)}$, without relying on potentially stringent content-style statistical independence (Kong et al., 2022; Xie et al., 2023; Shrestha & Fu, 2025) or sparse latent influence (Yan et al., 2023). Experiment results corroborate our identifiability theory.

**Limitations and Future Works.** In this work, we focus primarily on GAN-based implementations due to their explicit content-style generator structure and computational efficiency. However, GAN training can be unstable and may suffer from mode collapse. Modern diffusion and flow-matching models offer improved generation quality and diversity, making them attractive directions for extending the proposed framework. Incorporating the proposed content-style differential independence constraints into these models is nontrivial due to their fundamentally different architectures and training mechanisms. Extending the proposed CSDI framework to diffusion and flow-based generative models is an important direction for future work.

## Acknowledgment

This work was supported in part by the National Science Foundation (NSF) under Project ECCS-2450987, and in part by the NSF CAREER Award ECCS-2144889.

## Impact Statement

This paper presents work whose goal is to advance identifiability theory and scalable implementation of unsupervised generative model learning. There are many potential societal consequences of our work, none of which we feel must be specifically highlighted here.

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

# Supplementary Materials of "Content-Style Identification via Differential Independence"

## A. Invertibility of $\widehat{g}$

Let $\widehat{g}$ denote the learned generator mapping from the latent product manifold $\widehat{\mathcal{Z}} = \widehat{\mathcal{C}} \times \widehat{\mathcal{S}}$ to the data manifold $\mathcal{X}$. In this section we explains why, under correct latent dimensionality and regularity of the learned generator $\widehat{g}$. $\widehat{g}$ is bijective even though we do not explicitly impose an invertibility constraint on $\widehat{g}$. To that end, consider the following proposition.

**Proposition A.1** ((Sec. A.5. "Effects of the Uniformity Loss", Theorem 5 (Zimmermann et al., 2021)).). *Let $\mathcal{M}$ and $\mathcal{N}$ be simply connected and oriented $C^1$ manifolds without boundaries and $h : \mathcal{M} \to \mathcal{N}$ be a differentiable map. Further, let the random variable $z \in \mathcal{M}$ be distributed according to $z \sim p(z)$ for a regular function $p$, i.e., $0 < p < \infty$. If the push forward $h_\# p(z)$ of $p$ through $h$ is also a regular density, i.e., $0 < h_\# p < \infty$, then $h$ is a bijection.*

Proposition A.1 also applies when $\mathcal{M}$ and $\mathcal{N}$ are embedded in Euclidean spaces of different dimensions, as long as they have the same intrinsic manifold dimension (see Corollary 1 in (Zimmermann et al., 2021)). In our generative model, the data dimension $d$ may exceed $d_C + d_S$, this assumption is compatible with $\mathcal{X} \subseteq \mathbb{R}^d$ because the observations lie on a $(d_C + d_S)$-dimensional manifold embedded in $\mathbb{R}^d$.

Assume the latent dimensionalities $d_C$ and $d_S$ match the intrinsic dimensions of the true factors. For each domain $n \in [N]$, we apply Proposition A.1 with

$$\mathcal{M} = \widehat{\mathcal{Z}}^{(n)} = \widehat{\mathcal{C}} \times \widehat{\mathcal{S}}^{(n)}, \qquad \mathcal{N} = \mathcal{X}^{(n)}, \qquad h = \widehat{g}.$$

We assume $\widehat{\mathcal{Z}}^{(n)}$ and $\mathcal{X}^{(n)}$ are oriented $C^1$ manifolds without boundary.

We verify the assumptions of Proposition A.1 in our setting, step by step, as follows:

**(i) Simply connectedness of $\widehat{\mathcal{Z}}^{(n)}$.** Let $r_C \sim \mathcal{N}(0, I_{d_C})$ and $r_S^{(n)} \sim \mathcal{N}(0, I_{d_S})$ be random variables on $\mathbb{R}^{d_C}$ and $\mathbb{R}^{d_S}$, respectively. We learn $C^1$ invertible maps

$$e_C : \mathbb{R}^{d_C} \to \widehat{\mathcal{C}}, \qquad e_S^{(n)} : \mathbb{R}^{d_S} \to \widehat{\mathcal{S}}^{(n)},$$

and enforce their invertibility through (13). Under these assumptions, $\widehat{\mathcal{C}}$ is diffeomorphic to $\mathbb{R}^{d_C}$ and $\widehat{\mathcal{S}}^{(n)}$ is diffeomorphic to $\mathbb{R}^{d_S}$. Hence both are simply connected, and therefore their product

$$\widehat{\mathcal{Z}}^{(n)} = \widehat{\mathcal{C}} \times \widehat{\mathcal{S}}^{(n)}$$

is simply connected as well.

**(ii) Regularity of the latent density on $\widehat{\mathcal{Z}}^{(n)}$.** The Gaussian densities on $\mathbb{R}^{d_C}$ and $\mathbb{R}^{d_S}$ are strictly positive and finite everywhere. Since $e_C$ and $e_S^{(n)}$ are invertible and $C^1$, their pushforwards induce densities on $\widehat{\mathcal{C}}$ and $\widehat{\mathcal{S}}^{(n)}$ that are also regular. Consequently, the joint latent density $p_{\widehat{z}^{(n)}}$ on $\widehat{\mathcal{Z}}^{(n)}$ is regular:

$$0 < p_{\widehat{z}^{(n)}}(\widehat{z}^{(n)}) < \infty, \qquad \forall \widehat{z}^{(n)} \in \widehat{\mathcal{Z}}^{(n)}.$$

**(iii) Regularity of the induced data density on $\mathcal{X}^{(n)}$.** Let $p_{x^{(n)}}$ be the pushforward of $p_{\widehat{z}^{(n)}}$ through $\widehat{g}$:

$$p_{x^{(n)}} = \widehat{g}_\# p_{\widehat{z}^{(n)}}.$$

Since $p_{x^{(n)}}$ is also a regular density $0 < p_{x^{(n)}}(x) < \infty$, $\forall x \in \mathcal{X}^{(n)}$.

Steps (i)–(iii) verify all conditions of Proposition A.1. Therefore, for each domain $n \in [N]$, the learned generator $\widehat{g} : \widehat{\mathcal{Z}}^{(n)} \to \mathcal{X}^{(n)}$ is a bijection. In particular, the inverse $\widehat{f} = \widehat{g}^{-1}$ exists, and for each $x^{(n)} \in \mathcal{X}^{(n)}$,

$$\widehat{z}^{(n)} = \widehat{g}^{-1}(x^{(n)}) = \widehat{f}(x^{(n)}), \qquad \forall n \in [N]. \tag{17}$$

# B. Proof of Theorem 3.4

**Theorem 3.4** (Identifiability of Content and Style). *Assume that Eq. (1) and Assumption 3.3 hold. Let $\widehat{z} = (\widehat{c}, \widehat{s}^{(n)})$ be the latent components learned by solving (3). Then, the learned $\widehat{c}$ satisfies*

$$\widehat{c} = \gamma(c), \tag{5}$$

*where $\gamma : \mathcal{C} \to \widehat{\mathcal{C}}$ is an invertible mapping.*

*If, in addition, Assumption 3.1 holds and the style Jacobian satisfies $\mathrm{rank}(J_{s^{(n)}}g) = d_S$ , then for each domain $n$, the learned $\widehat{s}^{(n)}$ satisfies*

$$\widehat{s}^{(n)} = \delta(s^{(n)}), \qquad \forall\, n \in [N], \tag{6}$$

*where $\delta : \mathcal{S} \to \widehat{\mathcal{S}}$ is an invertible mapping.*

*Proof.* Let $\widehat{g}$ denote the learned bijective mapping from the latent product space $\widehat{\mathcal{Z}} = \widehat{\mathcal{C}} \times \widehat{\mathcal{S}}$ to the data manifold $\mathcal{X}$ by solving the problem in (3).

Let us denote the inverse mapping $\widehat{f} = \widehat{g}^{-1}$ and, for each sample $x^{(n)} \in \mathcal{X}$,

$$\widehat{z}^{(n)} = \widehat{g}^{-1}(x^{(n)}) = \widehat{f}(x^{(n)}) \quad \forall n \in [N]. \tag{18}$$

With the product structure $\widehat{z}^{(n)} = (\widehat{c}^{(n)}, \widehat{s}^{(n)})$, we denote the components by

$$\widehat{c}^{(n)} = \widehat{f}_C(x^{(n)}), \qquad \widehat{s}^{(n)} = \widehat{f}_S(x^{(n)}) \quad \forall n \in [N].$$

We will now prove this in following two steps:

**Step 1 (Content identifiability).** For content identifiability the proof follows the similar steps as in (Part C.1 Theorem 3.3 (Shrestha & Fu, 2025)).

**Proposition B.1** ((Summary of Part 1, Theorem 3.3 (Shrestha & Fu, 2025)).). *Under the nonlinear mixture model (1), suppose that Assumptions 3.3 hold and distribution matching constraint is satisfied in (3b). The learned content is identifiable up to an non-linear invertibile mapping of ground truth content $c$:*

$$\widehat{c} = \widehat{f}_C(x^{(n)}) = \gamma(c). \tag{19}$$

*for every domain $n \in [N]$.*

In particular, (19) implies that the extracted content $\widehat{f}_C(x^{(n)})$ depends only on the shared variable $c$ and is invariant to the domain-specific style variable $s^{(n)}$.

Next we will use this result along with differential independence to identify the style.

**Step 2 (Style identifiability).**

For each domain $n$, we learn the following style mapping

$$\widehat{s}^{(n)} := \widehat{f}_S\big(g(c, s^{(n)})\big) = \delta(c, s^{(n)}),$$

and the learned content is

$$\widehat{c} = \widehat{f}_C\big(g(c, s^{(n)})\big) = \gamma(c).$$

We have the following:

$$g(c, s^{(n)}) = \widehat{g}\big(\widehat{c} = \gamma(c), \ \widehat{s}^{(n)} = \delta(c, s^{(n)})\big), \tag{20}$$

where $\gamma(c)$ is obtained from the content-identifiability result.

Taking the Jacobian of both sides of (20) with respect to $s^{(n)}$ and applying the multivariate chain rule gives

$$J_{s^{(n)}} g(c, s^{(n)}) = J_{\widehat{s}^{(n)}} \widehat{g}\big(\gamma(c), \delta(c, s^{(n)})\big) J_{s^{(n)}} \delta(c, s^{(n)}), \tag{21}$$

since $\boldsymbol{J}_{\boldsymbol{s}^{(n)}}\gamma(\boldsymbol{c}) = 0$ (the learned content $\widehat{\boldsymbol{c}} = \gamma(\boldsymbol{c})$ is independent of $\boldsymbol{s}^{(n)}$).

From (21) and the property $\mathcal{R}(\boldsymbol{AB}) \subseteq \mathcal{R}(\boldsymbol{A})$, we have

$$\mathcal{R}\Big(\boldsymbol{J}_{\boldsymbol{s}^{(n)}}\boldsymbol{g}(\boldsymbol{c},\boldsymbol{s}^{(n)})\Big) \subseteq \mathcal{R}\Big(\boldsymbol{J}_{\widehat{\boldsymbol{s}}^{(n)}}\widehat{\boldsymbol{g}}(\gamma(\boldsymbol{c}),\boldsymbol{\delta}(\boldsymbol{c},\boldsymbol{s}^{(n)}))\Big).$$

From the assumption that $\boldsymbol{J}_{\boldsymbol{s}^{(n)}}\boldsymbol{g}$ has full column rank $d_S$ and $\mathrm{rank}(\boldsymbol{J}_{\widehat{\boldsymbol{s}}^{(n)}}\widehat{\boldsymbol{g}}\,\boldsymbol{J}_{\boldsymbol{s}^{(n)}}\boldsymbol{\delta}) = \min\{\mathrm{rank}(\boldsymbol{J}_{\widehat{\boldsymbol{s}}^{(n)}}),\mathrm{rank}(\boldsymbol{J}_{\boldsymbol{s}^{(n)}}\boldsymbol{\delta})\} = d_S$. Since the left-hand side equals $d_S$, both factors must satisfy

$$\mathrm{rank}\Big(\boldsymbol{J}_{\widehat{\boldsymbol{s}}}\widehat{\boldsymbol{g}}(\gamma(\boldsymbol{c}),\boldsymbol{\delta}(\boldsymbol{c},\boldsymbol{s}^{(n)}))\Big) = d_{\mathrm{S}}, \qquad \mathrm{rank}\Big(\boldsymbol{J}_{\boldsymbol{s}^{(n)}}\boldsymbol{\delta}(\boldsymbol{c},\boldsymbol{s}^{(n)})\Big) = d_{\mathrm{S}}. \tag{22}$$

Hence the above inclusion becomes an equality:

$$\mathcal{R}\Big(\boldsymbol{J}_{\boldsymbol{s}^{(n)}}\boldsymbol{g}(\boldsymbol{c},\boldsymbol{s}^{(n)})\Big) = \mathcal{R}\Big(\boldsymbol{J}_{\widehat{\boldsymbol{s}}^{(n)}}\widehat{\boldsymbol{g}}(\gamma(\boldsymbol{c}),\boldsymbol{\delta}(\boldsymbol{c},\boldsymbol{s}^{(n)}))\Big). \tag{23}$$

Taking the Jacobian of both sides of (20) with respect to $\boldsymbol{c}$ and applying the multivariate chain rule gives,

$$\boldsymbol{J}_{\boldsymbol{c}}\boldsymbol{g}(\boldsymbol{c},\boldsymbol{s}^{(n)}) = \boldsymbol{J}_{\widehat{\boldsymbol{s}}^{(n)}}\widehat{\boldsymbol{g}}(\gamma(\boldsymbol{c}),\boldsymbol{\delta}(\boldsymbol{c},\boldsymbol{s}^{(n)}))\boldsymbol{J}_{\boldsymbol{c}}\boldsymbol{\delta}(\boldsymbol{c},\boldsymbol{s}^{(n)}) + \boldsymbol{J}_{\widehat{\boldsymbol{c}}}\widehat{\boldsymbol{g}}(\gamma(\boldsymbol{c}),\boldsymbol{\delta}(\boldsymbol{c},\boldsymbol{s}^{(n)}))\boldsymbol{J}_{\boldsymbol{c}}\gamma(\boldsymbol{c}). \tag{24}$$

Enforcing the differential-independence regularization in (3c) we get:

$$\Big(\boldsymbol{J}_{\widehat{\boldsymbol{s}}^{(n)}}\widehat{\boldsymbol{g}}(\widehat{\boldsymbol{c}},\widehat{\boldsymbol{s}}^{(n)})\Big)^{\top}\boldsymbol{J}_{\widehat{\boldsymbol{c}}}\widehat{\boldsymbol{g}}(\widehat{\boldsymbol{c}},\widehat{\boldsymbol{s}}) \; = \; \boldsymbol{0} \qquad \text{for all } (\widehat{\boldsymbol{c}},\widehat{\boldsymbol{s}}) \in \widehat{\mathcal{C}} \times \widehat{\mathcal{S}}. \tag{25}$$

Let $P_{\widehat{\mathcal{S}}}(\cdot)$ denote the orthogonal projector onto $\mathcal{R}\Big(\boldsymbol{J}_{\widehat{\boldsymbol{s}}}\widehat{\boldsymbol{g}}(\gamma(\boldsymbol{c}),\boldsymbol{\delta}(\boldsymbol{c},\boldsymbol{s}^{(n)}))\Big)$. By Assumption 3.1 (style directions are orthogonal to content directions in the true generator), we have

$$P_{\widehat{\mathcal{S}}}\Big(\boldsymbol{J}_{\boldsymbol{c}}\boldsymbol{g}(\boldsymbol{c},\boldsymbol{s}^{(n)})\Big) = \boldsymbol{0}, \tag{26}$$

because (23) identifies the learned style tangent space with the true style tangent space. Moreover, by (25), the range of $\boldsymbol{J}_{\widehat{\boldsymbol{c}}}\widehat{\boldsymbol{g}}$ is orthogonal to the range of $\boldsymbol{J}_{\widehat{\boldsymbol{s}}}\widehat{\boldsymbol{g}}$, hence

$$P_{\widehat{\mathcal{S}}}\Big(\boldsymbol{J}_{\widehat{\boldsymbol{c}}}\widehat{\boldsymbol{g}}(\gamma(\boldsymbol{c}),\boldsymbol{\delta}(\boldsymbol{c},\boldsymbol{s}^{(n)}))\,\boldsymbol{J}_{\boldsymbol{c}}\gamma(\boldsymbol{c})\Big) = \boldsymbol{0}. \tag{27}$$

Applying $P_{\widehat{\mathcal{S}}}$ to both sides of (24) and using (26)–(27) yields

$$P_{\widehat{\mathcal{S}}}\Big(\boldsymbol{J}_{\widehat{\boldsymbol{s}}}\widehat{\boldsymbol{g}}(\gamma(\boldsymbol{c}),\boldsymbol{\delta}(\boldsymbol{c},\boldsymbol{s}^{(n)}))\,\boldsymbol{J}_{\boldsymbol{c}}\boldsymbol{\delta}(\boldsymbol{c},\boldsymbol{s}^{(n)})\Big) \; = \; \boldsymbol{0}. \tag{28}$$

Since the columns of $\boldsymbol{J}_{\widehat{\boldsymbol{s}}}\widehat{\boldsymbol{g}}(\cdot)$ lie in $\mathcal{R}(\boldsymbol{J}_{\widehat{\boldsymbol{s}}}\widehat{\boldsymbol{g}}(\cdot))$ by definition, the projector acts as the identity on this term, and (28) simplifies to

$$\boldsymbol{J}_{\widehat{\boldsymbol{s}}}\widehat{\boldsymbol{g}}(\gamma(\boldsymbol{c}),\boldsymbol{\delta}(\boldsymbol{c},\boldsymbol{s}^{(n)}))\,\boldsymbol{J}_{\boldsymbol{c}}\boldsymbol{\delta}(\boldsymbol{c},\boldsymbol{s}^{(n)}) \; = \; \boldsymbol{0}. \tag{29}$$

By (22), $\boldsymbol{J}_{\widehat{\boldsymbol{s}}}\widehat{\boldsymbol{g}}(\cdot)$ has full column rank $d_S$; therefore the only solution is

$$\boldsymbol{J}_{\boldsymbol{c}}\boldsymbol{\delta}(\boldsymbol{c},\boldsymbol{s}^{(n)}) \; = \; \boldsymbol{0}, \tag{30}$$

i.e., $\boldsymbol{\delta}(\boldsymbol{c},\boldsymbol{s}^{(n)})$ is independent of $\boldsymbol{c}$. Hence there exists a map $\boldsymbol{\delta}: \mathcal{S} \to \widehat{\mathcal{S}}$ such that

$$\boldsymbol{\delta}(\boldsymbol{c},\boldsymbol{s}^{(n)}) \; = \; \boldsymbol{\delta}(\boldsymbol{s}^{(n)}), \qquad \text{and thus} \qquad \widehat{\boldsymbol{s}}^{(n)} \; = \; \boldsymbol{\delta}(\boldsymbol{s}^{(n)}). \tag{31}$$

Since $\boldsymbol{\delta} = \widehat{\boldsymbol{f}} \circ \boldsymbol{g}$ is an invertible function and $\mathcal{S}^{(n)}$ is a simply connected set. The learned style code is an invertible and content-independent transformation of the true style variable:

$$\widehat{\boldsymbol{s}}^{(n)} \; = \; \boldsymbol{\delta}(\boldsymbol{s}^{(n)}).$$

This completes the proof.

$\square$

## C. Proof of Theorem 3.6

**Theorem 3.6** (Robustness of Identifiability). *Assume that Assumption 3.5 holds and that the style Jacobian satisfies* $\mathrm{rank}(\boldsymbol{J}_{\boldsymbol{s}^{(n)}}\boldsymbol{g}) = d_S$. *Then, the following statements hold:*

*(a)* $\widehat{\boldsymbol{c}} = \boldsymbol{\gamma}(\boldsymbol{c})$, *where* $\boldsymbol{\gamma} : \mathcal{C} \to \widehat{\mathcal{C}}$ *is an invertible mapping;*

*(b)* $\widehat{\boldsymbol{s}}^{(n)} = \boldsymbol{\delta}(\boldsymbol{c}, \boldsymbol{s}^{(n)})$, *where* $\boldsymbol{\delta} : \mathcal{C} \times \mathcal{S}^{(n)} \to \widehat{\mathcal{S}}$ *is the induced learned-style map, and for every domain* $n \in [N]$, *the learned* $\widehat{\boldsymbol{s}}^{(n)}$ *satisfies*

$$\left\| \boldsymbol{J}_{\boldsymbol{c}}\widehat{\boldsymbol{s}}^{(n)} \right\|_2 \leq \tag{7}$$
$$\sin\xi \big/ \sigma_{\min}\!\Big( \boldsymbol{J}_{\widehat{\boldsymbol{s}}}\widehat{g}\big(\widehat{\boldsymbol{c}},\widehat{\boldsymbol{s}}^{(n)}\big)\Big) \, \| \boldsymbol{J}_{\boldsymbol{c}}\boldsymbol{g}(\boldsymbol{c},\boldsymbol{s}^{(n)}) \|_2.$$

*Proof.* (a) For content identifiability the proof follows the similar steps as in Step 1 in Proof of Theorem 3.4.

(b) Now we establish the robustness of style identifiability when tangent subspaces of content and style are not exactly orthogonal to each other. We start from the Assumption 3.5 where subspaces are approximately orthogonal in the sense that their smallest principal angle satisfies

$$\angle\big(\mathcal{R}(\boldsymbol{J}_{\boldsymbol{c}}\boldsymbol{g}(\boldsymbol{c},\boldsymbol{s}^{(n)})),\ \mathcal{R}(\boldsymbol{J}_{\boldsymbol{s}^{(n)}}\boldsymbol{g}(\boldsymbol{c},\boldsymbol{s}^{(n)})) \big) \geq \min\{\frac{\pi}{2} \pm \xi\} = \frac{\pi}{2} - \xi,$$

for some small $\xi \geq 0$.

Let $P_{\mathcal{C}}$ and $P_{\mathcal{S}}$ denote the orthogonal projection matrices onto $\mathcal{C} = \mathcal{R}(\boldsymbol{J}_{\boldsymbol{c}}\boldsymbol{g}(\boldsymbol{c},\boldsymbol{s}^{(n)}))$, $\mathcal{S} = \mathcal{R}(\boldsymbol{J}_{\boldsymbol{s}^{(n)}}\boldsymbol{g}(\boldsymbol{c},\boldsymbol{s}^{(n)}))$.

Now, we start from the chain-rule identity obtained in (24) in the proof of style identifiability as,

$$\boldsymbol{J}_{\boldsymbol{c}}\boldsymbol{g}(\boldsymbol{c},\boldsymbol{s}^{(n)}) = \boldsymbol{J}_{\widehat{\boldsymbol{s}}}\widehat{g}\big(\boldsymbol{\gamma}(\boldsymbol{c}),\boldsymbol{\delta}(\boldsymbol{c},\boldsymbol{s}^{(n)})\big)\,\boldsymbol{J}_{\boldsymbol{c}}\boldsymbol{\delta}(\boldsymbol{c},\boldsymbol{s}^{(n)}) + \boldsymbol{J}_{\widehat{\boldsymbol{c}}}\widehat{g}\big(\boldsymbol{\gamma}(\boldsymbol{c}),\boldsymbol{\delta}(\boldsymbol{c},\boldsymbol{s}^{(n)})\big)\,\boldsymbol{J}_{\boldsymbol{c}}\boldsymbol{\gamma}(\boldsymbol{c}).$$

Projecting onto the learned style subspace $\widehat{\mathcal{S}}$ and using the orthogonality of the learned decoder gives

$$\boldsymbol{J}_{\widehat{\boldsymbol{s}}}\widehat{g}\big(\boldsymbol{\gamma}(\boldsymbol{c}),\boldsymbol{\delta}(\boldsymbol{c},\boldsymbol{s}^{(n)})\big)\,\boldsymbol{J}_{\boldsymbol{c}}\boldsymbol{\delta}(\boldsymbol{c},\boldsymbol{s}^{(n)}) = P_{\widehat{\mathcal{S}}}\Big(\boldsymbol{J}_{\boldsymbol{c}}\boldsymbol{g}(\boldsymbol{c},\boldsymbol{s}^{(n)})\Big).$$

From the equality of style subspaces established in (23),

$$P_{\widehat{\mathcal{S}}}\Big(\boldsymbol{J}_{\boldsymbol{c}}\boldsymbol{g}(\boldsymbol{c},\boldsymbol{s}^{(n)})\Big) = P_{\mathcal{S}}\Big(\boldsymbol{J}_{\boldsymbol{c}}\boldsymbol{g}(\boldsymbol{c},\boldsymbol{s}^{(n)})\Big).$$

Since the columns of $\boldsymbol{J}_{\boldsymbol{c}}\boldsymbol{g}(\boldsymbol{c},\boldsymbol{s}^{(n)})$ lie in $\mathcal{R}(\boldsymbol{J}_{\boldsymbol{c}}\boldsymbol{g}(\boldsymbol{c},\boldsymbol{s}^{(n)}))$, we can write

$$\boldsymbol{J}_{\boldsymbol{c}}\boldsymbol{g}(\boldsymbol{c},\boldsymbol{s}^{(n)}) = P_{\mathcal{C}}\Big(\boldsymbol{J}_{\boldsymbol{c}}\boldsymbol{g}(\boldsymbol{c},\boldsymbol{s}^{(n)})\Big),$$

and hence

$$P_{\mathcal{S}}\Big(\boldsymbol{J}_{\boldsymbol{c}}\boldsymbol{g}(\boldsymbol{c},\boldsymbol{s}^{(n)})\Big) = P_{\mathcal{S}}P_{\mathcal{C}}\Big(\boldsymbol{J}_{\boldsymbol{c}}\boldsymbol{g}(\boldsymbol{c},\boldsymbol{s}^{(n)})\Big).$$

The assumption $\angle\big(\mathcal{R}(\boldsymbol{J}_{\boldsymbol{c}}\boldsymbol{g}(\boldsymbol{c},\boldsymbol{s}^{(n)})),\ \mathcal{R}(\boldsymbol{J}_{\boldsymbol{s}^{(n)}}\boldsymbol{g}(\boldsymbol{c},\boldsymbol{s}^{(n)}))\big) \geq \frac{\pi}{2} - \xi$ is equivalent to saying that the maximum possible overlap between a unit vector in $\mathcal{C}$ and a unit vector in $\mathcal{S}$ is at most $\sin\xi$. In terms of orthogonal projections, this can be written as

$$\|P_{\mathcal{S}}P_{\mathcal{C}}\|_{\mathrm{op}} = \|P_{\mathcal{C}}P_{\mathcal{S}}\|_{\mathrm{op}} \leq \cos(\frac{\pi}{2} - \xi) = \sin\xi.$$

where $\|.\|_{\mathrm{op}}$ is the operator norm. Therefore,

$$\left\| \boldsymbol{J}_{\widehat{\boldsymbol{s}}}\widehat{g}\big(\boldsymbol{\gamma}(\boldsymbol{c}),\boldsymbol{\delta}(\boldsymbol{c},\boldsymbol{s}^{(n)})\big)\,\boldsymbol{J}_{\boldsymbol{c}}\boldsymbol{\delta}(\boldsymbol{c},\boldsymbol{s}^{(n)}) \right\|_2 = \left\| P_{\mathcal{S}}P_{\mathcal{C}}\big(\boldsymbol{J}_{\boldsymbol{c}}\boldsymbol{g}(\boldsymbol{c},\boldsymbol{s}^{(n)})\big) \right\|_2$$
$$\leq \|P_{\mathcal{S}}P_{\mathcal{C}}\|_{\mathrm{op}}\,\|\boldsymbol{J}_{\boldsymbol{c}}\boldsymbol{g}(\boldsymbol{c},\boldsymbol{s}^{(n)})\|_2$$
$$\leq \sin\xi\,\|\boldsymbol{J}_{\boldsymbol{c}}\boldsymbol{g}(\boldsymbol{c},\boldsymbol{s}^{(n)})\|_2.$$

Because $J_{\widehat{s}}\widehat{g}(\gamma(c), \delta(c, s^{(n)}))$ has full column rank, its smallest singular value is strictly positive, and thus

$$\left\| J_{\widehat{s}}\widehat{g}(\gamma(c), \delta(c, s^{(n)})) \, J_c\delta(c, s^{(n)}) \right\|_2 \geq \sigma_{\min}\left( J_{\widehat{s}}\widehat{g}(\gamma(c), \delta(c, s^{(n)})) \right) \left\| J_c\delta(c, s^{(n)}) \right\|_2.$$

Combining the above two bounds yields

$$\left\| J_c\delta(c, s^{(n)}) \right\|_2 \leq \frac{\sin \xi}{\sigma_{\min}\left( J_{\widehat{s}}\widehat{g}(\gamma(c), \delta(c, s^{(n)})) \right)} \left\| J_c g(c, s^{(n)}) \right\|_2.$$

This completes the proof.

$\square$

## D. Implementation Details

### D.1. Jacobian Regularization

#### D.1.1. PROPERTIES OF THE PROPOSED $\mathcal{L}_{\text{orth}}$ IN (14)

**Scale invariance.** The proposed differential-independence regularizer $\mathcal{L}_{\text{orth}}$ is (approximately) scale-invariant. Ignoring the small constant $\epsilon > 0$ and using the Cauchy–Schwarz inequality, we have

$$\|J_{s^{(n)}}^\top J_c\|_F^2 \leq \|J_{s^{(n)}}\|_F^2 \|J_c\|_F^2, \tag{32}$$

which implies

$$\frac{\|J_{s^{(n)}}^\top J_c\|_F^2}{\|J_c\|_F^2 \|J_{s^{(n)}}\|_F^2} \leq 1, \tag{33}$$

whenever $\|J_c\|_F, \|J_{s^{(n)}}\|_F > 0$. This normalization keeps the regularizer on a bounded scale and makes it less sensitive to the overall magnitudes of the Jacobians, which helps prevent the term from dominating the objective due to uniform rescaling of the latent parameterization.

**Non-trivial solutions.** The denominator $\|J_{\widehat{c}}\|_F^2 \|J_{\widehat{s}^{(n)}}\|_F^2 + \epsilon$ in $\mathcal{L}_{\text{orth}}$ avoids the trivial solution where the loss is reduced by making the Jacobians nearly zero. Since, the distribution-matching objective in (11) implicitly encourages $\widehat{g}$ to be bijective (see Appendix A), which further discourages degenerate (collapsed) content and style representations. For small $\epsilon$, the loss is roughly unchanged if both Jacobians are scaled down together, so simply shrinking $J_{\widehat{c}}$ and $J_{\widehat{s}^{(n)}}$ does not decrease $\mathcal{L}_{\text{orth}}$.

#### D.1.2. STOCHASTIC APPROXIMATION OF DIFFERENTIAL INDEPENDENCE LOSS

For each data sample we sample the noise probes

$$v \in \mathbb{R}^d \sim \mathcal{N}(\mathbf{0}, I_d)$$

such that $E[vv^\top] = I_d$. We then compute the Jacobian vector product for content $c$ and $s$ using vector jacobian product (VJP) as follows:

$$J_c^\top v = \nabla_c \langle g(c, s), v \rangle \in \mathbb{R}^{d_C}, \qquad J_s^\top v = \nabla_s \langle g(c, s), v \rangle \in \mathbb{R}^{d_S}.$$

We then estimate our differential independence regularization as follows:

$$\frac{\|J_{s^{(n)}}^\top J_c\|_F^2}{\|J_c\|_F^2 \|J_{s^{(n)}}\|_F^2 + \epsilon} = \frac{\|\mathbb{E}_v[J_{s^{(n)}}^\top v(J_c^\top v)^\top]\|_F^2}{\mathbb{E}_v\left[\|J_c^\top v\|_2^2\right] \mathbb{E}_v\left[\|J_{s^{(n)}}^\top v\|_2^2\right] + \epsilon} \tag{34}$$

The equivalence in (34) can be shown easily. For the numerator,

$$\|\mathbb{E}_v[J_{s^{(n)}}^\top v(J_c^\top v)^\top]\|_F^2 = \|\mathbb{E}_v[J_{s^{(n)}}^\top vv^\top J_c]\|_F^2 = \|J_{s^{(n)}}^\top \mathbb{E}_v[vv^\top]J_c\|_F^2 = \|J_{s^{(n)}}^\top J_c\|_F^2.$$

Similarly for the normalization terms,

$$\|J_c^\top v\|_2^2 = (J_c^\top v)^\top (J_c^\top v) = v^\top J_c J_c^\top v.$$

Since a scalar equals its trace,

$$\boldsymbol{v}^\top \boldsymbol{J_c}\boldsymbol{J_c}^\top \boldsymbol{v} = \mathrm{tr}(\boldsymbol{v}^\top \boldsymbol{J_c}\boldsymbol{J_c}^\top \boldsymbol{v}) = \mathrm{tr}(\boldsymbol{J_c}\boldsymbol{J_c}^\top \boldsymbol{v}\boldsymbol{v}^\top),$$

where we used cyclic invariance of the trace. Taking expectation and using linearity,

$$\begin{aligned}
\mathbb{E}_{\boldsymbol{v}}[\boldsymbol{v}^\top \boldsymbol{J_c}\boldsymbol{J_c}^\top \boldsymbol{v}] &= \mathrm{tr}\big(\boldsymbol{J_c}\boldsymbol{J_c}^\top \, \mathbb{E}_{\boldsymbol{v}}[\boldsymbol{v}\boldsymbol{v}^\top]\big) \\
&= \mathrm{tr}(\boldsymbol{J_c}\boldsymbol{J_c}^\top) = \mathrm{tr}(\boldsymbol{J_c}^\top \boldsymbol{J_c}) \\
&= \|\boldsymbol{J_c}\|_F^2.
\end{aligned}$$

Similarly, we can write $\|\boldsymbol{J}_{\boldsymbol{s}^{(n)}}^\top \boldsymbol{v}\|_2^2 = \|\boldsymbol{J}_{\boldsymbol{s}^{(n)}}\|_F^2$.

### D.1.3. COMPUTATIONAL COMPLEXITY FOR JACOBIAN COMPUTATION

We want to learn a generator $\boldsymbol{g} : \mathbb{R}^{d_C} \times \mathbb{R}^{d_S} \to \mathbb{R}^d$ where $d$ is the dimension of data and $d \gg d_C, d_S$ although the data and latent lie on the same low-dimensional manifold. Denote the Jacobians

$$\boldsymbol{J_c} := \frac{\partial \boldsymbol{g}}{\partial \boldsymbol{c}} \in \mathbb{R}^{d \times d_C}, \qquad \boldsymbol{J_s} := \frac{\partial \boldsymbol{g}}{\partial \boldsymbol{s}} \in \mathbb{R}^{d \times d_S}.$$

**Explicit Jacobian computation.** For each sample, materializing $\boldsymbol{J_c} \in \mathbb{R}^{d \times d_C}$ and $\boldsymbol{J_s} \in \mathbb{R}^{d \times d_S}$ requires storing $d(d_C + d_S)$ entries, hence for a batch:

$$\mathrm{Memory}_{\mathrm{explicit}} = \mathcal{O}\big(B\, d(d_C + d_S)\big).$$

Using reverse-mode differentiation, forming a full Jacobian requires $\mathcal{O}(d)$ backward passes per sample (one per output coordinate), so the batch FLOPs are

$$\mathrm{FLOPs}_{\mathrm{explicit}} = \mathcal{O}\big(B\, d\, F_{\mathrm{bwd}}\big) \text{ to obtain } \boldsymbol{J_c}, \boldsymbol{J_s} \text{ (up to constants).}$$

Given explicit Jacobians, the cross-term $\boldsymbol{J_s}^\top \boldsymbol{J_c} \in \mathbb{R}^{d_S \times d_C}$ costs a matrix multiply $(d_S \times d) \cdot (d \times d_C)$ per sample, hence

$$\mathrm{FLOPs}\big(\boldsymbol{J_s}^\top \boldsymbol{J_c}\big) = \mathcal{O}\big(B\, d\, d_S d_C\big).$$

**Noise probing via VJP (ours).**

For each probe $\boldsymbol{v} \in \mathbb{R}^d$ with $\boldsymbol{E}[\boldsymbol{v}\boldsymbol{v}^\top] = I$, we compute $\boldsymbol{J_c}^\top \boldsymbol{v}$ and $\boldsymbol{J_s}^\top \boldsymbol{v}$ via one backward pass of the scalar $\langle \boldsymbol{g}(\boldsymbol{c}, \boldsymbol{s}), \boldsymbol{v} \rangle$. With $K$ probes per sample, batch FLOPs are

$$\mathrm{FLOPs}_{\mathrm{probe}} = \mathcal{O}\big(B\, K\, F_{\mathrm{bwd}}\big).$$

Forming the outer products $(\boldsymbol{J_s}^\top \boldsymbol{v})(\boldsymbol{J_c}^\top \boldsymbol{v})^\top \in \mathbb{R}^{d_S \times d_C}$ adds $\mathcal{O}(B\, K\, d_S d_C)$ FLOPs, typically negligible relative to backprop through $\boldsymbol{g}$, hence

$$\mathrm{FLOPs}_{\mathrm{probe}} = \mathcal{O}\big(B\, K\, F_{\mathrm{bwd}} + B\, K\, d_S d_C\big) \approx \mathcal{O}\big(B\, K\, F_{\mathrm{bwd}}\big).$$

The additional memory we need is

$$\mathrm{Memory}_{\mathrm{probe}} = \mathcal{O}\big(B\, d + B\, d_S d_C\big),$$

since we store the probe vectors in $\mathbb{R}^d$ and an accumulator in $\mathbb{R}^{d_S \times d_C}$, but never store $d \times d_C$ or $d \times d_S$ Jacobians.

Explicit Jacobians scale as $\mathcal{O}(B\, d(d_C + d_S))$ memory and $\mathcal{O}(B\, d)$ backward passes, whereas noise probing replaces the $\mathcal{O}(d)$ factor by $\mathcal{O}(K)$ (with $K \ll d$) and reduces the memory by $(d_C + d_S)$ times.

## E. Experiment Details

### E.1. Computational Resources for Experiments

All of our experiments are conducted on a machine with $1\times$ NVIDIA V100 (32GB VRAM) Intel(R) Xeon(R) Platinum 8168 CPU @ 2.70GHz, 1.5 TB RAM.

*Table 2.* RGB prototype pools used for digit color ($\mathcal{D}$) and background color ($\mathcal{B}$).

| $\mathcal{D}$ | $\mathcal{B}$ | Name (informal) | RGB $(R, G, B)$ |
|---|---|---|---|
| $d_0$ | $b_3$ | red-ish | (255, 50, 50) |
| $d_1$ | $b_2$ | blue-ish | (50, 50, 255) |
| $d_2$ | $b_1$ | green-ish | (50, 255, 50) |
| $d_3$ | $b_0$ | yellow-ish | (255, 255, 50) |
| $d_4$ | $b_9$ | magenta-ish | (255, 50, 255) |
| $d_5$ | $b_8$ | cyan-ish | (50, 255, 255) |
| $d_6$ | $b_7$ | orange-ish | (255, 140, 50) |
| $d_7$ | $b_6$ | purple-ish | (140, 50, 255) |
| $d_8$ | $b_5$ | teal | (0, 170, 170) |
| $d_9$ | $b_4$ | olive/mustard | (180, 160, 20) |

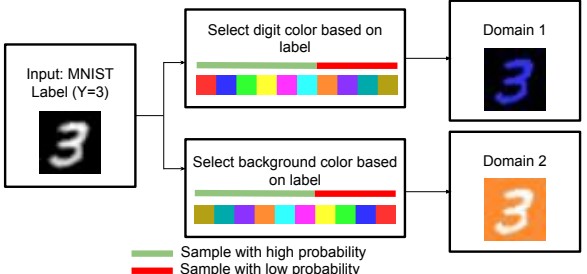

*Figure 9.* High-level construction of two-domain MNIST with dependent content and style. From a grayscale MNIST digit (content $Y = 3$), we generate two domains by applying domain-specific styles: colored digits (Domain 1) and colored digits (Domain 2). Style variables are sampled in a label-dependent (biased) manner to induce controlled content-style dependence.

## E.2. Two-domain (Digit Color + Background Color) MNIST Experiments

We create a two-domain variant of MNIST in which the *content* is the digit identity, while the *style* is domain-specific and *label-dependent*: color of digit in Domain 1 and color of background in Domain 2. Starting from the MNIST training set $\{(x_i, y_i)\}_{i=1}^N$ with $y_i \in \{0, \ldots, 9\}$ and grayscale images $x_i$, we resize each image to $32 \times 32$ and normalize intensities to $x_i \in [0, 1]^{32 \times 32}$. Each domains contain 60,000 MNIST digit images. The high level construction of dependent MNIST is shown in Fig. 9.

**Style candidate pools.** To generate the colored digit, we define a discrete pool of 10 RGB prototypes $\mathcal{D} = \{d_0, \ldots, d_9\}$. For colored background, we do the same and fix 10 RGB prototypes $\mathcal{B} = \{b_0, \ldots, b_9\}$. The color pools $\mathcal{D}$ and $\mathcal{B}$ were constructed using the colors as listed in Table 2.

**Label-dependent allowed sets.** To inject a controlled label–style dependence, we assign to each label $y$ an *allowed* subset of colors by removing three consecutive candidates (with wrap-around and indices modulo 10 for colors):

$$\mathcal{D}_y = \mathcal{D} \setminus \{d_y, d_{y+1}, d_{y+2}\}, \qquad \mathcal{B}_y = \{b_0, \ldots, b_9\} \setminus \{b_y, b_{y+1}, b_{y+2}\}.$$

Hence, for every $y$, the allowed-set sizes satisfy $|\mathcal{B}_y| = |\mathcal{D}_y| = 7$.

**Domain construction (content shared, style differs).** We generate two domains from the same underlying digits:

- **Domain 1 (Colored digits).** For each $(x_i, y_i)$, we sample a digit color $D_i$ using the label-dependent rule below and map the digit color to RGB prototype $D_i$, i.e. $\boldsymbol{x}_i^{(1)} = \text{Colorize Digit}(x_i; D_i)$.

- **Domain 2 (Colored background).** For each $(x_i, y_i)$, we sample a background color $B_i$ using the label-dependent rule below and set $\boldsymbol{x}_i^{(2)} = \text{Colorize Background}(x_i; B_i)$ (by mapping the background color of $x_i$ to RGB prototype $B_i$).

Thus, the two domains share the same digit identity (content), while the style is digit color in Domain 1 and background color in Domain 2, both correlated with the label.

**Label-dependent (biased) sampling.** Given a digit label $Y = y$, we draw a digit color $D$ and a background color $B$ from mixtures that favor the allowed sets $\mathcal{D}_y$ and $\mathcal{B}_y$ of the *true* label:

$$D \mid (Y = y) \sim p_{\text{digit}} \cdot \text{Unif}(\mathcal{D}_y) + (1 - p_{\text{digit}}) \cdot \frac{1}{9} \sum_{j \neq y} \text{Unif}(\mathcal{D}_j),$$

$$B \mid (Y = y) \sim p_{\text{bg}} \cdot \text{Unif}(\mathcal{C}_y) + (1 - p_{\text{bg}}) \cdot \frac{1}{9} \sum_{j \neq y} \text{Unif}(\mathcal{B}_j),$$

with $p_{\text{digit}} = p_{\text{bg}} = 0.8$. Equivalently, with probability $p_{\text{digit}}$ (resp. $p_{\text{bg}}$) we sample uniformly from the allowed set associated with the true label; otherwise we first choose an *incorrect* label $j \neq y$ uniformly from the remaining 9 labels and

*Table 3.* Generator $\widehat{\boldsymbol{g}}$ for MNIST experiment.

| Block | Specification |
|---|---|
| Linear | Concatenate $(\boldsymbol{c}, \boldsymbol{s})$ and map to $128 \times 2 \times 2$ |
| BN, Up | Upsample to $4 \times 4$ |
| Conv, BN, lReLU | 128 filters, $3 \times 3$, stride 1, padding 1 |
| Up, Conv, BN, lReLU | 128 filters, $3 \times 3$, stride 1, padding 1 (to $8 \times 8$) |
| Up, Conv, BN, lReLU | 64 filters, $3 \times 3$, stride 1, padding 1 (to $16 \times 16$) |
| Up, Conv, BN, lReLU | 32 filters, $3 \times 3$, stride 1, padding 1 (to $32 \times 32$) |
| Conv, Tanh | 3 filters, $3 \times 3$, stride 1, padding 1 |

*Table 4.* Discriminator $\widehat{\boldsymbol{d}}^{(n)}$ for MNIST experiment.

| Block | Specification |
|---|---|
| Label embedding | Map domain label to a $1 \times 32 \times 32$ spatial map |
| Concatenate | Concatenate image and embedding $\Rightarrow 4 \times 32 \times 32$ input |
| Conv, lReLU, Drop | 64 filters, $3 \times 3$, stride 2, padding 1 |
| Conv, lReLU, Drop, BN | 128 filters, $3 \times 3$, stride 2, padding 1 |
| Conv, lReLU, Drop, BN | 256 filters, $3 \times 3$, stride 2, padding 1 |
| Linear, Sigmoid | Fully-connected head to a scalar output |

then sample uniformly from $\mathcal{A}_j$ (resp. $\mathcal{C}_j$). In our implementation, the two style variables are conditionally independent given the label, i.e., $D \perp B \mid Y$.

This procedure induces a controlled correlation between content and style: the digit label $Y$ becomes predictive of both digit color $D$ and background color $B$ through the label-specific allowed sets, with the bias strength governed by $(p_{\text{digit}}, p_{\text{bg}})$, while the underlying digit shape (content) remains unchanged.

**Evaluations.** To measure the quality of generated images, we use FID scores (Heusel et al., 2017); the lower the FID score is, the better. In addition, to measure the style diversity of generations, we use the LPIPS distance (Zhang et al., 2018) between pairs of images generated with the same content $\widehat{\boldsymbol{c}}$ but different styles $\widehat{\boldsymbol{s}}^{(n)}$. We report the average over calculated LPIPS distances; the larger the LPIPS score is, the more diverse the images are (in terms of style).

**Neural network architectures.** We adopt DCGAN architectures tailored to $32 \times 32$ MNIST-style images. The content encoder $\boldsymbol{e}_C : \mathbb{R}^{d_C} \to \mathbb{R}^{d_C}$, style encoders $\boldsymbol{e}_S^{(n)} : \mathbb{R}^{d_S} \to \mathbb{R}^{d_S}$ and decoders $\boldsymbol{t}_C$ and $\boldsymbol{t}_S^{(n)}$ are linear layers. The generator and discriminator backbones are summarized in Table 3 and Table 4 respectively. Here, Conv denotes a convolution, BN means batch normalization, lReLU means LeakyReLU with slope 0.2, Up means nearest-neighbor upsampling by factor 2, and Drop means Dropout with probability 0.25.

**Training configuration.** For both Block Independence GAN (B.I. GAN) (Shrestha & Fu, 2025) and our CSDI-GAN, we train generator $\widehat{\boldsymbol{g}}$ for 500 epochs with batch size 128 using Adam (learning rate $10^{-4}$, $\beta_1 = 0.5$, $\beta_2 = 0.999$). We share 32 dimensional noise among content and style, i.e. $d_{C_1} = d_{C_2} = 48$ and $d_{S_1} = 16$, resulting in $d_C = 96$ and $d_S = 32$ for statistical dependency modeling of content and style. We set the hyperparameters of regularization term to be $\lambda_{\text{inv}} = 0.001$ and $\lambda_{\text{orth}} = 1.0$. For number of probes we set $K = 8$ and $\epsilon = 10^{-8}$.

**Domain translation procedure.** Given a trained generator $\widehat{\boldsymbol{g}}$ and trained style encoders $\boldsymbol{e}_S^{(n)}$, domain translation is done via two-step process. In the first step, we solve for $(\widehat{\boldsymbol{c}}, \widehat{\boldsymbol{s}}^{(i)}) \triangleq \arg\min_{\tilde{\boldsymbol{c}}, \tilde{\boldsymbol{s}}^{(i)}} \text{div}(\widehat{\boldsymbol{g}}(\tilde{\boldsymbol{c}}, \tilde{\boldsymbol{s}}^{(i)}), \boldsymbol{x}^{(i)})$, where $\text{div}(\cdot, \cdot)$ is a divergence measure, to obtain the content-style representation of an image $\boldsymbol{x}^{(i)}$. This step is usually referred to as GAN inversion (Xia et al., 2023)), which can be implemented via many approaches. Following (Shrestha & Fu, 2025), we use a pre-trained VGG-16 convolutional neural network (Simonyan & Zisserman, 2015) for $\text{div}(\cdot, \cdot)$, and use Adam optimizer to solve this step. We similarly extract the style $\widehat{\boldsymbol{s}}^{(t)}$ from a target domain image. In the second step, we combine the extracted content $\widehat{\boldsymbol{c}}$ from the source image with the target style $\widehat{\boldsymbol{s}}^{(t)}$ to create the translated $\widehat{\boldsymbol{x}}^{(t)} = \widehat{\boldsymbol{g}}(\widehat{\boldsymbol{c}}, \widehat{\boldsymbol{s}}^{(t)})$.

**Additional results.** We provide additional qualitative results for MNIST generation and translation tasks. Fig. 10 compares generated images from B.I. GAN (Shrestha & Fu, 2025), as well as from CSDI-GAN with and without $\mathcal{L}_{\text{orth}}$. Our proposal CSDI-GAN with $\mathcal{L}_{\text{orth}}$ and B.I. GAN can generate *diverse styles* while keeping the *content fixed*, indicating that distribution matching helps capture content structure. However, when we *fix style* and *vary content*, B.I. GAN fails to preserve the intended disentanglement, producing digits whose content remains coupled with style.

Notably, CSDI-GAN without the content-style differential independence regularizer $\mathcal{L}_{\text{orth}}$ fails in both scenarios—it cannot

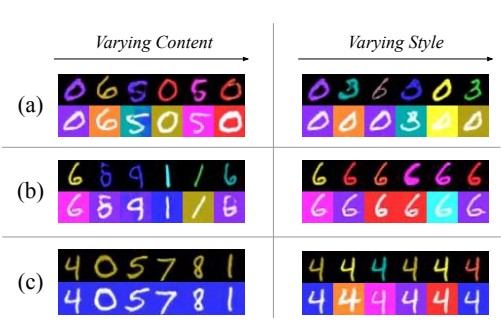

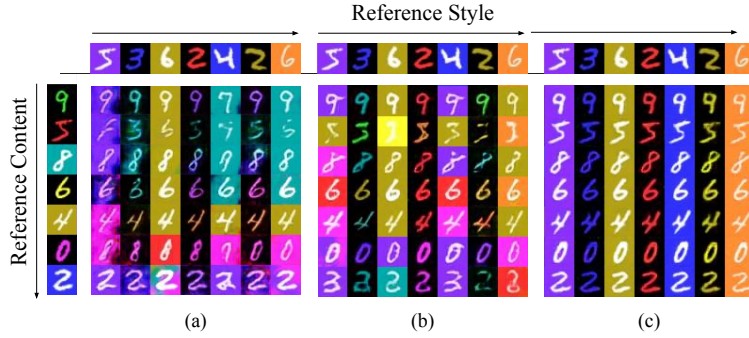

*Figure 10.* Generation task on two-domain MNIST with [Left] same $\widehat{s}^{(n)}$ and varying $\widehat{c}$, and [Right] same $\widehat{c}$, varying $\widehat{s}^{(n)}$: (a) CSDI-GAN w/o $\mathcal{L}_{\text{orth}}$ (FID: 27.45, LPIPS: 0.2674); (b) B.I. GAN (FID: 26.25, LPIPS: 0.2685); (c) CSDI-GAN w/ $\mathcal{L}_{\text{orth}}$ (**FID: 23.80**, **LPIPS: 0.3321**).

*Figure 11.* Translation task on two-domain MNIST data: (a) CSDI-GAN w/o $\mathcal{L}_{\text{orth}}$, (b) B.I. GAN, (c) Proposed – CSDI-GAN w/ $\mathcal{L}_{\text{orth}}$‘.

*Table 5.* Disentanglement evaluation on MNIST using the $R^2$ score between ground-truth and learned content/style representations.

| Method | $R^2$ Content | $R^2$ Style (averaged across domains) |
|---|---|---|
| CSDI w/o $\mathcal{L}_{\text{orth}}$ | 0.7324 | 0.5618 |
| BI | 0.7744 | 0.7338 |
| CSDI | 0.7675 | **0.8995** |

consistently disentangle the content and style information. This illustrates the importance of our key regularizer $\mathcal{L}_{\text{orth}}$ to enforce Assumption 3.1.

Fig. 11 reports additional translation results, where each row corresponds to the same content and each column corresponds to the same style. Our method CSDI-GAN (with $\mathcal{L}_{\text{orth}}$) yields better-behaved latent factors and enables consistent translation across both axes. In contrast, B.I. GAN and the unregularized variant of CSDI-GAN exhibit noticeable content-style entanglement and often fail to translate reliably. Overall, the proposed content-style differential independence regularization provides the most consistent and successful translations.

**Disentanglement Evaluation.** Ground-truth content $c$ (digit identity) and style $s^{(n)}$ (background color or digit color) are available in MNIST, we directly evaluate disentanglement using the $R^2$ score. We train a linear regressor to predict $c$ from $\widehat{c}$ and $s^{(n)}$ from $\widehat{s}^{(n)}$. As shown in Table 5, CSDI achieves the highest style $R^2$ score while maintaining competitive content prediction performance. This suggests that the orthogonality constraint is crucial for accurately learning disentangled content and style representations.

**Parameter Sensitivity Analysis.**

Each loss term in the objective (11) is designed to promote the identifiability of the ground-truth latent variables, as characterized by our theoretical analysis. We further examine the sensitivity of the proposed method on MNIST dataset to the hyperparameters parameters used in our objectives. In all experiments, we keep all other hyperparameters fixed at their default values and vary only the parameter under study.

Table 6 shows the sensitivity with respect to $\lambda_{\text{inv}}$ and $\lambda_{\text{orth}}$. The results indicate that the model is relatively stable over a moderate range of values. Setting either regularization weight to zero degrades the FID, confirming the importance of the corresponding loss terms. Very large regularization weights can also hurt performance, suggesting that overly strong constraints may limit the flexibility of the learned representation. In practice, the FID on the target domain can be used as a simple criterion for selecting these parameters.

We also evaluate the sensitivity to the Jacobian-regularizer parameters $K$ and $\epsilon$. As shown in Table 7, performance is generally best for moderate parameter choices. Increasing $K$ beyond a moderate value does not yield consistent gains while increasing computational cost. Similarly, the method is stable for small values of $\epsilon$, whereas an overly large value of $\epsilon$ degrades the FID.

*Table 6.* Sensitivity analysis of the loss weights $\lambda_{\text{inv}}$ and $\lambda_{\text{orth}}$ on the MNIST dataset.

| $\lambda_{\text{inv}}$ | FID | LPIPS | $\lambda_{\text{orth}}$ | FID | LPIPS |
|---|---|---|---|---|---|
| 0 | 28.5 | 0.30 | 0 | 27.4 | 0.27 |
| 0.001 | 23.8 | 0.33 | 0.01 | 23.6 | 0.33 |
| 1.0 | 24.1 | 0.32 | 1.0 | 23.8 | 0.33 |
| 10.0 | 30.2 | 0.21 | 10.0 | 31.5 | 0.31 |

*Table 7.* Sensitivity analysis of the Jacobian-regularizer parameters $K$ and $\epsilon$ on the MNIST dataset.

| $K$ | FID | LPIPS | $\epsilon$ | FID | LPIPS |
|---|---|---|---|---|---|
| 2 | 26.1 | 0.32 | $10^{-18}$ | 23.6 | 0.34 |
| 4 | 24.5 | 0.33 | $10^{-12}$ | 23.4 | 0.33 |
| 8 | 23.8 | 0.33 | $10^{-8}$ | 23.8 | 0.33 |
| 16 | 23.9 | 0.33 | $10^{-4}$ | 24.3 | 0.32 |
| 32 | 23.7 | 0.34 | 0.1 | 27.1 | 0.32 |

*Table 8.* Computational overhead comparison on AFHQ over 720K iterations.

| Method | Training time (hours) | Peak GPU memory (GB) |
|---|---|---|
| StyleGAN | 23.11 | 9.38 |
| BI-GAN | 28.32 | 12.65 |
| I-GAN | 28.90 | 11.30 |
| CSDI-GAN | 30.09 | 14.20 |

### E.3. AFHQ and CelebA-HQ Experiments

**Datasets.** We conduct experiments on **AFHQ** (Choi et al., 2020) and **CelebA-HQ** (Karras et al., 2018) for both multi-domain image generation and image-to-image translation. **AFHQ** contains animal-face images from 3 domains (*cat*, *dog*, and *wild*) with 5153/4739/4738 training images and 500/500/500 test images, respectively. **CelebA-HQ** consists of high-resolution human-face images with semantic attributes (e.g., gender, hair color, expressions, and so on). We construct 2 domains based on gender, where the male and female domains contain 17,943 and 10,057 training images, and 1000 images in each domain for testing. For both datasets, we resize all images to $128 \times 128$ for training and evaluation.

**Neural network architecture.** We implement the generator $\widehat{g}$ and discriminator $\widehat{d}^{(n)}$ following the StyleGAN2-ADA design (Karras et al., 2020b). The StyleGAN2-ADA generator consists of a mapping network and a synthesis network. In our implementation, we represent $\widehat{g}$ using only the *synthesis* network and discard the mapping network, similar to (Shrestha & Fu, 2025). Instead of a mapping network, we learn a content encoder $e_C$ and a collection of style encoders $\{e_S^{(n)}\}_{n=1}^{N}$. Each encoder is a 3-layer MLP with 512 hidden units in each hidden layer.

We set $d_{C_1} = 6$ and $d_{C_2} = 494$, so the input dimension to $e_C$ is $d_C = d_{C_1} + d_{C_2} = 500$. For each style encoder $e_S^{(n)}$, we set $d_{S_1} = 6$ and define the input dimension as $d_S = d_{S_1} + d_{C_1} = 12$. To condition the synthesis network, we feed the output of $e_C$ into the first five layers of the synthesis network, and inject the outputs of $\{e_S^{(n)}\}_{n=1}^{N}$ into the remaining layers of $\widehat{g}$.

We construct the networks $t_C$ and $\{t_S^{(n)}\}_{n=1}^{N}$ to mirror $e_C$ and $\{e_S^{(n)}\}_{n=1}^{N}$, respectively, using the same MLP architecture but with the input and output dimensions swapped.

**Training hyperparameters.** Our GAN training setup largely follows the StyleGAN2-ADA recipe (Karras et al., 2020b). We optimize all networks using Adam (Kingma & Ba, 2015) with learning rate $2.5 \times 10^{-3}$ and batch size 32, and set the momentum parameters to $\beta_1 = 0$ and $\beta_2 = 0.99$. Across all datasets, we run training for 200,000 iterations. We set the hyperparameters of regularization term to be $\lambda_{\text{inv}} = 0.001$ and $\lambda_{\text{orth}} = 2.0$. For number of probes we set $K = 4$ and $\epsilon = 10^{-8}$.

**Computational cost.** We analyze the computational overhead of CSDI-GAN relative to GAN-based baselines. We report the wall-clock training time and peak GPU memory usage in Table 8 for all methods over 720K iterations on AFHQ using a single V100 GPU. As shown in Table 8, CSDI-GAN incurs a moderate increase in training time and memory usage compared to the baselines. This overhead mainly comes from the Jacobian regularization used to encourage content-style disentanglement. Importantly, this additional cost is accompanied by improved FID and LPIPS performance, suggesting a favorable trade-off between computational cost and generation quality.

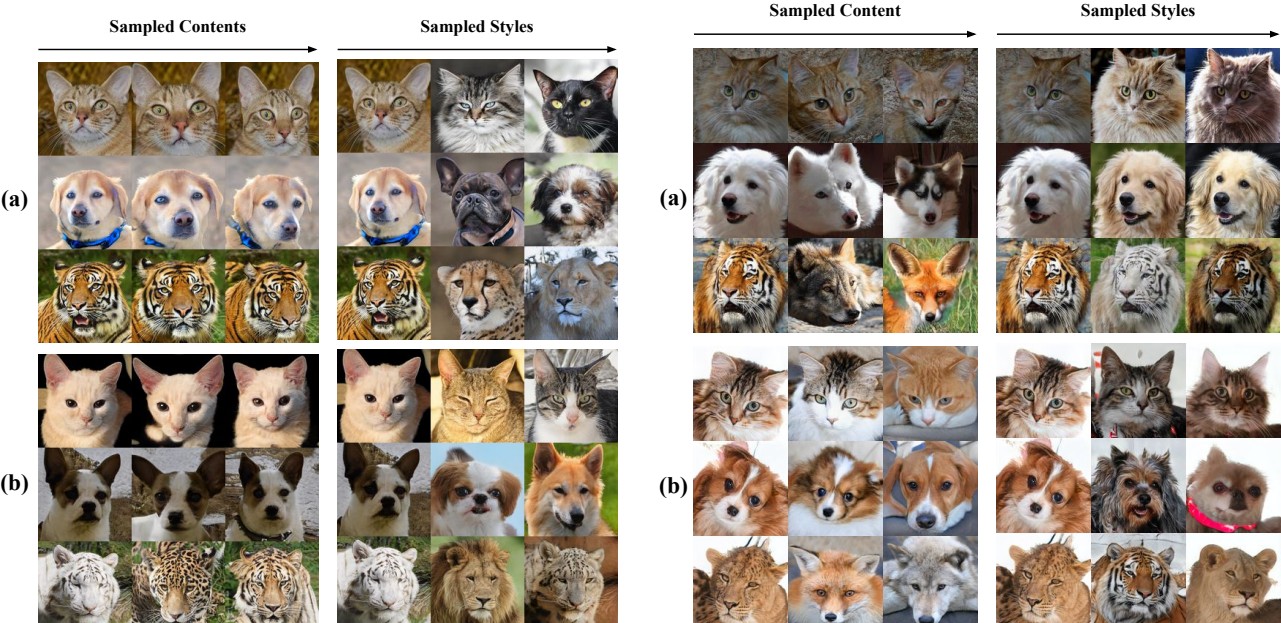

*Figure 12.* Generation task with AFHQ dataset. (a) Our CSDI-GAN with $\mathcal{L}_{\text{orth}}$ (b) B.I. GAN (Shrestha & Fu, 2025).

*Figure 13.* Generation task with AFHQ dataset. (a) CSDI-GAN w/o $\mathcal{L}_{\text{orth}}$ (b) I-StyleGAN (Xie et al., 2023).

### E.3.1. MULTI-DOMAIN DATA GENERATION

We provide additional qualitative results for multi-domain generation. Fig. 12 compares generated images from CSDI-GAN and B.I. GAN (Shrestha & Fu, 2025), where each row is intended to share the same content and each column the same style. CSDI-GAN produces visually cleaner samples and preserves the desired structure: it can maintain *consistent style* while *varying content*, and likewise maintain *consistent content* while *varying style*. In contrast, B.I. GAN often fails to keep the sampled style consistent across different contents, leading to noticeable style drift and imperfect disentanglement.

Fig. 13 further compares (a) CSDI-GAN without $\mathcal{L}_{\text{orth}}$ and (b) I-StyleGAN (Xie et al., 2023). Notably, removing our content-style differential independence regularizer $\mathcal{L}_{\text{orth}}$ degrades generation in both axes, indicating that the learned latents no longer reliably separate content from style. Overall, these results highlight the importance of $\mathcal{L}_{\text{orth}}$ in stabilizing multi-domain generation and enforcing Assumption 3.1.

Similarly, Fig. 14 compares generated images from CSDI-GAN and B.I. GAN (Shrestha & Fu, 2025) under the same evaluation protocol (rows share content and columns share style). CSDI-GAN better preserves the intended disentanglement, exhibiting clearer and more consistent changes when varying content or style.

Fig. 15 further reports multi-domain generation results for (a) I-StyleGAN (Xie et al., 2023) and (b) CSDI-GAN without $\mathcal{L}_{\text{orth}}$. Overall, our full model yields more distinguishable and stable content/style variations, while the competing methods also produce reasonable results but with less consistent separation between the two factors.

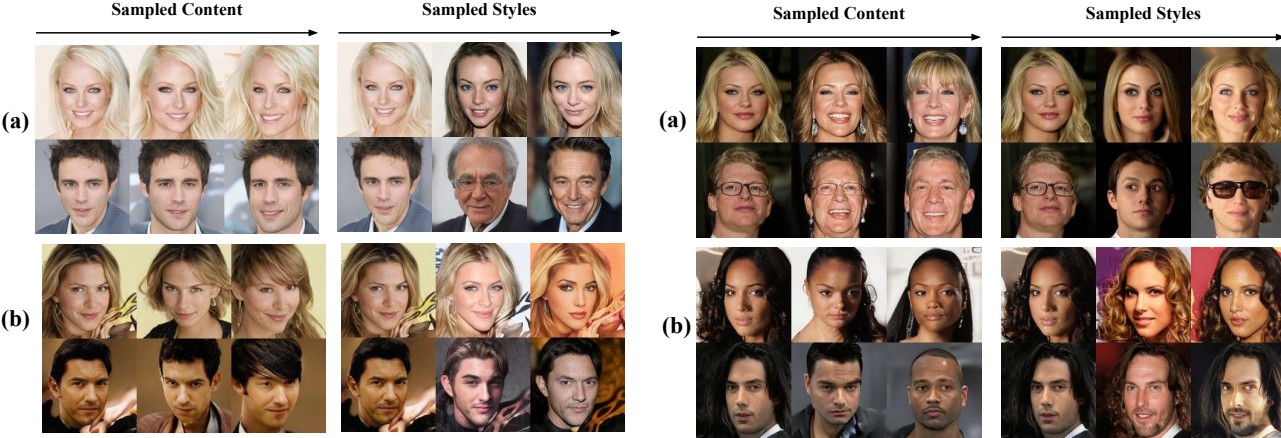

*Figure 14.* Generation task with CelebA-HQ dataset. (a) Our CSDI-GAN with $\mathcal{L}_{\mathrm{orth}}$ (b) B.I. GAN (Shrestha & Fu, 2025).

*Figure 15.* Generation task with CelebA-HQ dataset. (a) I-StyleGAN (Xie et al., 2023) (b) CSDI-GAN w/o $\mathcal{L}_{\mathrm{orth}}$.

