# OpenReview forum: "Content-Style Identification via Differential Independence"
_ICML.cc/2026/Conference — ICML 2026 regular_

### Official Review · Reviewer_xgMt · 2026-03-09

**Soundness:** 2
**Presentation:** 2
**Significance:** 1
**Originality:** 3
**Overall Recommendation:** 2
**Confidence:** 4

**Summary:**

The authors address the challenge of unpaired multi-domain learning, where cross-domain correspondences are not observed. They argue that while content (shared information) and style (domain-specific information) may be statistically dependent in reality (e.g., lighting depending on object geometry), they should induce orthogonal variations on the data manifold.To operationalize this, they introduce a Jacobian-based orthogonality regularizer (L_orth) for generative models. Because computing full Jacobians for high-dimensional data is prohibitive, they develop a scalable stochastic approximation using Hutchinson's trace estimation and Vector-Jacobian Products (VJP). They provide a theoretical proof of identifiability for both content and style under these conditions and demonstrate superior performance on image generation and translation tasks using MNIST, AFHQ, and CelebA-HQ datasets.

**Compliance With Llm Reviewing Policy:**

Affirmed.

**Key Questions For Authors:**

1. How does the degree of "orthogonality violation" in the ground-truth data manifold affect the error bounds of your identifiability result?

2. The Jacobian regularizer includes an \epsilon term and requires K noise probes. How sensitive are the FID and LPIPS results to the choice of K and the magnitude of \epsilon across different data modalities?

**Limitations:**

yes.

**Strengths And Weaknesses:**

Strengths

1. Unlike prior work, this method does not require content and style to be statistically independent, making it more applicable to real-world scenarios where these factors naturally correlate.Scalability to High Dimensions: The use of noise-probing VJP estimators allows the model to handle high-resolution images by avoiding the explicit formation of large Jacobian matrices, reducing memory requirements by a factor of roughly d_C + d_S.

2. The paper provides a rigorous identifiability theorem (Theorem 3.4) showing that content and style can be recovered up to invertible transformations without relying on sparse Jacobian assumptions.Empirical Robustness: Experiments across multiple datasets show that the model (CSDI-GAN) more consistently preserves style and content during counterfactual generation compared to existing baselines like I-StyleGAN and B.I. GAN.


Weaknesses

The authors acknowledge that the assumption of exact mutual orthogonality may not always hold in practice, which could impact the model's reliability in certain domains.Complex Implementation: While scalable, the framework involves multiple loss terms (L_GAN, L_inv, L_orth) and requires careful numerical stabilization (e.g., the \epsilon constant in the regularizer). The current implementation is primarily focused on GAN architectures; its effectiveness in newer generative frameworks like Diffusion or Flow Matching remains largely unexplored in this paper.

---

> ### Author Rebuttal · Authors · 2026-03-31
>
> We thank the reviewer for their comments. We hope to stress that the main contribution of this work is theoretical: we introduce **content-style differential independence** and establish identifiability of content and style under this condition. Our implementation is designed mainly to validate the theory and show that the proposed objective is practically optimizable.
>
> **[Orthogonality Assumption Violation]**
>
> To address Jacobian orthogonality violation, we derive a new theoretical result showing that when the content and style subspaces are not exactly orthogonal, one obtains an **approximate identifiability guarantee**, with an error bound that depends on the degree of orthogonality violation.
>
> Specifically, assume the principal angle between the content and style tangent subspaces is at least $\frac{\pi}{2}-\xi$. Then
>
> $$\|| J_{c}\hat{s}^{(n)}\||_2 \le \frac{\sin \xi}{\sigma\_{\min} (J\_{\hat{s}} \hat{g}(\hat{c},\hat{s}^{(n)}) ) } \||J\_{c}x^{(n)}\||_2. $$
>
> Exact style identifiability is recovered when $\xi=0$, while for small $\xi>0$, the learned style $\hat{s}^{(n)}$ is approximately content-independent.
>
> Due to space limits, we refer the reviewer to our *reply to Reviewer YeXN for the theorem statement and proof sketch*.
>
> To illustrate this, we conducted a simulation where the degree of orthogonality violation is controlled by $\gamma$, with $J_s^T J_c = 0$ when $ \gamma = 0$, otherwise not. We then measured the R2-score between the recovered and ground-truth style variables over 5 trials by fitting a linear regressor to predict $s^{(n)}$ from $\hat s^{(n)}$.
> The $R^2$ score is highest near $\gamma =0$, and quite robust under violation of orthogonality. This is consistent with our theory. We will include both the error bound and this sensitivity analysis in the revised paper.
>
> *Data generation process:*
> - $c \in \mathbb{R}^2: VonMises(\mu=2.5, \kappa=2.0), s^{(1)} \in \mathbb{R}^2 : Beta(\alpha =3, \beta = 1) + 0.2 * tanh(c) . s^{(2)} \in \mathbb{R}^2 : Gamma(shape=0.5, scale=3.0) + 0.2 * tanh(c)$
> - $X^{(n)} = g(c,s^{(n)}) = [u(c) + v(s);  u(c) - v(s)] + \gamma [c^2; s^2] \in \mathbb{R}^4$, where $u(c)=c+0.1c^3,   v(s)=s+0.1s^3.$
>
> | $\gamma $ | $R^2$-score style |
> | :--- | :---: |
> | -100 | 0.4435 ± 0.0520 |
> | -5 | 0.6787 ± 0.0320 |
> | -2 | 0.6723 ± 0.0223 |
> | 0 | 0.7399 ± 0.0037 |
> | 2 | 0.7275 ± 0.0421 |
> | 5 | 0.6311 ± 0.0455 |
> | 100 | 0.4328 ± 0.0311 |
>
> **[Different Generative Model Implementation]**
>
> We agree that our implementation is focused on GAN-based architecture, and an exciting direction is to implement our CSDI criterion in newer generative models such as diffusion/flow. Indeed, in principle, our identifiability result of CSDI is agnostic to different implementations. But as score-based methods do not have an explicit generator like x = g(c, s), the implementation details and training losses may require substantial re-design to realize our identifiable formulation in Eq. 3. Such an extension to diffusion/flow models is an interesting but nontrivial open problem, which we leave for future work and had explicitly mentioned as a limitation in our paper.
>
> **[Implementation & Parameter Sensitivity Analysis]**
>
> We would like to clarify that each loss term serves the purpose to identify the ground truth latents as proved in our theorem. We hope to stress that tuning the parameters is not very cumbersome. In practice, using FID of the target domain can assist tuning them. The model itself is also not very sensitive to the parameters. In the sensitivity analysis, we keep all other hyperparameters fixed at their default values and vary only the parameter under study.  The numerical results are reported below for MNIST Dataset.
>
> | $\lambda_{inv}$ | LPIPS | FID | &nbsp; | $\lambda_{orth}$ | LPIPS | FID |
> | :--- | :---: | :---: | :---: | :--- | :---: | :---: |
> | 0 | 28.5 | 0.30 | &#124; | 0 | 27.4 | 0.27 |
> | 0.001 | 23.8 | 0.33 | &#124; | 0.01 | 23.6 | 0.33 |
> | 1.0 | 24.1 | 0.32 | &#124; | 1.0 | 23.8 | 0.33 |
> | 10.0 | 30.2 | 0.21 | &#124; | 10.0 | 31.5 | 0.31 |
>
> We also evaluated sensitivity to the Jacobian-regularizer parameters $K$ and $\epsilon$. The results indicate that performance depends on these choices, with the best results typically attained for moderate values rather than at the extremes. In particular, larger $K$ increases computational cost without yielding consistent gains, and overly large $\epsilon$ can degrade performance.
>
> | $K$ | FID | LPIPS | &nbsp; | $\epsilon$ | FID | LPIPS |
> | :--- | :---: | :---: | :---: | :--- | :---: | :---: |
> | 2 | 26.1 | 0.32 | &#124; | 1e-18 | 23.6 | 0.34 |
> | 4 | 24.5 | 0.33 | &#124; | 1e-12 | 23.4 | 0.33 |
> | 8 | 23.8 | 0.33 | &#124; | 1e-8 | 23.8 | 0.33 |
> | 16 | 23.9 | 0.33 | &#124; | 1e-4 | 24.3 | 0.32 |
> | 32 | 23.7 | 0.34 | &#124; | 0.1 | 27.1 | 0.32 |
>
> We will include these analysis in the revised paper.

---

> > ### Author Rebuttal · Reviewer_xgMt · 2026-04-06
> >
> > The overall contribution of the work is undermined by the use of outdated datasets and baseline techniques. To demonstrate the paper's value in the current landscape, evaluations against state-of-the-art methods on contemporary benchmarks are necessary.

---

> > > ### Author Response · Authors · 2026-04-06
> > >
> > > We respectfully disagree with the reviewer’s **new comment**, which was not raised in the original review and has only appeared after the rebuttal.
> > >
> > > As noted in our rebuttal, our key contribution is in laying the theoretical foundation for content-style learning under an alternative assumption. The baselines we used are the most recent and relevant for the identifiability problem under consideration and the datasets used are also standard and used for the validation of the considered identifiability theory.
> > >
> > > Given the generic nature of the comment, we wonder if the reviewer could specify which datasets and baselines in the work are outdated, and why? We would be happy to discuss further if the reviewer can clarify their raised concern.

---

### Official Review · Reviewer_YV2e · 2026-03-12

**Soundness:** 3
**Presentation:** 3
**Significance:** 3
**Originality:** 3
**Overall Recommendation:** 5
**Confidence:** 4

**Summary:**

This paper proposes a new content-style identification theory which is built upon orthogonal tangent subspace assumption, followed by a corresponding GAN-based identification algorithm, named CSDI-GAN. Experiments on synthetic and real-world image datasets show clear advantage of CSDI-GAN over existing baselines, in terms of counterfactual generation quality and diversity.

**Compliance With Llm Reviewing Policy:**

Affirmed.

**Final Justification:**

The proposed CSDI-GAN algorithm is a clear contribution to the community, which enjoys stronger identification performance and theoretical guarantee. Though the differential independence assumption may have no guarantee in practice, the experimental results provide support for its validity.  During rebuttal, the authors promise to add discussions on their assumptions and other existing assumptions, after which I suggest a clear acceptance.

**Key Questions For Authors:**

1.	Is CSDI assumption strictly weaker than statistical independence assumption, or just a substitute?
2.	Does CSDI-GAN algorithm require the knowledge of dimensionality of content and style variables? How do you set the dimensionality in your implementation?

**Limitations:**

yes

**Strengths And Weaknesses:**

Contributions of this paper are two parts: the identifiability theory under differential independence, and the identification algorithm CSDI-GAN, both of which are novel to the best of my knowledge.

For the identifiability theory, the CSDI assumption has its merit that it differs from previous statistical independence or Jacobian sparsity assumption. This provides a new option for selecting a proper assumption set. One drawback here is that the authors neither make a comparison between these assumptions, nor point out some typical situation that CSDI assumption will hold in practice. This hinders the potential application of proposed theory. By the way, there have been contemporary works considering substitute assumptions for statistical independence. I find that CSDI assumption looks similar to the **mechanistic independence** in [this paper](https://openreview.net/forum?id=0VVdai71xb). Though not counted as one weakness, I suggest the authors to compare their assumptions with such literatures in the next version.

For the identification algorithm, I am glad to see the proposed CSDI-GAN achieves better identification performance. This provides a new tool for image translation, and may further be extended to solve robustness problems.

---

> ### Author Rebuttal · Authors · 2026-03-31
>
> We thank reviewers for their careful reading and valuable comments.
>
> **[Comparing CSDI with other assumptions]**
>
> We agree that the paper should better position CSDI relative to statistical independence and Jacobian sparsity. Our point is not that CSDI strictly subsumes these assumptions, but that it offers a different modeling alternative. Statistical independence *constrains the joint distribution of content and style,* and Jacobian sparsity *constrains coordinate-wise influence of the latent variables on observed data*. In contrast, CSDI *constrains the local geometry of the data manifold, requiring content- and style-induced local variations to lie in orthogonal tangent subspaces*. We also note that CSDI assumption is useful in settings where content and style may not be statistically independent [R1, R2, R3], and where the generator Jacobian may be dense [R4] but their local effects on data remain geometrically separable.
>
> **[Situations where CSDI holds]**
>
> CSDI assumption typically holds where observations are formed by decoupled generative mechanisms. For example, consider 3D rendering, where object geometry (content) and illumination (style) interact to produce an image. Mechanistically, an intervention on illumination (style) produces a pixel-shift that is perpendicular to the structural pixel-shift produced by altering the object's mesh (content). While these factors may be correlated in natural dataset (e.g., certain object shapes might only be observed under specific lighting), CSDI ensures the generative mechanisms of content and style remain decoupled, allowing content-style identification from observed data.
> Furthermore, successful generative models such as VAE [R5] and StyleGAN2 [R6] implicitly promote a similar concept of Jacobian orthogonality, and they’re also known to induce disentangled representations. These empirical results imply that such orthogonal Jacobian regularizations help infer ground-truth latent variables, indicating such orthogonality models hold to a certain extent. We refer to Remark 3.2 for a discussion between these empirical works and our results.
>
> **[CSDI vs Mechanistic Independence]**
>
> We agree that CSDI is related in spirit to mechanistic independence, in that both move beyond statistical independence and instead characterize disentanglement through how latent factors act on the observations rather than through the latent density. Mechanistic Independence describes how latent factors sparsely influence data coordinates, which allows disentanglement. Specifically, this class of conditions describe the sparsity pattern in Jacobian/Hessian of the data generating function. While CSDI also uses Jacobian to characterize the data generating function, we assume orthogonality between the content Jacobian block and style Jacobian block. We will discuss the relation between Mechanistic Independence and CSDI in the revised version.
>
> **[CSDI vs Statistical Independence]**
>
> Our CSDI assumption should be viewed primarily as an alternative structural assumption, rather than as a strict weakening of statistical independence in the logical sense. In general, statistical independence and differential independence do not imply one another, so neither assumption strictly contains the other. However, CSDI is weaker in an important practical sense: it does not require content and style to be statistically independent as random variables, and therefore remains applicable in settings where content and style are statistically dependent, which is common in real data [R1, R2, R3]. Instead of requiring global distributional independence, CSDI only constrains the local geometric interaction between content- and style-induced variations on the data manifold. We will clarify this distinction more explicitly in the paper.
>
> **[Knowledge of dimensionality]**
>
> Theoretically, CSDI-GAN assumes that the dimensionalities of the content and style variables are known and correctly specified. In practice, these dimensions are treated as hyperparameters, and in our implementation we set them manually, following common practice in prior work [R7, R8].
>
> **References**
> [R1] von Kügelgen et al., 2021, Self supervised learning with data augmentations provably isolates content from style
>
> [R2] Schölkopf et al., 2021, Toward causal representation learning
>
> [R3] Yan et al., 2023, Counterfactual generation with identifiability guarantees
>
> [R4] Nguyen et. al 2025, Diverse influence component analysis: A geometric approach to nonlinear mixture identifiability
>
> [R5] Rolinek et al., 2019, Variational autoencoders pursue PCA directions
>
> [R6] Karras et al., 2020, Analyzing and improving the image quality of StyleGAN
>
> [R7] Xie et al., 2023, Multi-domain image generation and translation with identifiability guarantees
>
> [R8] Shrestha & Fu, 2025, Content-style learning from un- aligned domains: Identifiability under unknown latent dimensions

---

> > ### Author Rebuttal · Reviewer_YV2e · 2026-04-01
> >
> > The contribution of this paper is clear. I decide to maintain my positive score.

---

### Official Review · Reviewer_YeXN · 2026-03-13

**Soundness:** 3
**Presentation:** 2
**Significance:** 3
**Originality:** 3
**Overall Recommendation:** 4
**Confidence:** 3

**Summary:**

The paper uses differential independence to identify content and style factors in generative models. The method enforces orthogonality between content and style variations on the data manifold, which disentanglement even when factors are (or could be) statistically dependent. This approach also outperforms existing methods in counterfactual generation and domain translation on high-resolution datasets.

**Compliance With Llm Reviewing Policy:**

Affirmed.

**Final Justification:**

Th authors addressed my questions on domain generalization and decomposition, with theoretical proof. I will maintain my positive score.

**Key Questions For Authors:**

I merged questions into weaknesses.

**Limitations:**

yes

**Strengths And Weaknesses:**

Strengths:
1. Using hutchinson’s trace estimator is an efficeint way to calculate the orthogonal directions.
2. The theoretical constraints are relaxed compared to the previous works, which give this work more practical meanings.

Weakness&questions:
1. The model works well at unpaired data, but the domain variability assumption requires at least two domains with sufficiently diverse distributions to function. It's unclear that what happens when the differences of two domains is entangled with each other, for example, disentangle ID and a cartoon pixar styles.
2. In L083, the authors define the generator g as a smooth and bijective mapping, but real-world image data often contains sharp transitions or discrete structures (like the edge of an object moving against a background), where the proposition A.1 might not hold.
3. The authors model the tangent space of the data manifold as a direct sum of content and style subspaces (L132 right), this assumes that all the possible variation in the data can be decomposed into either content or style. but if there are features belongs to both manifold, will the orthogonality constraint L_orth force the model to collapse certain features or create artifacts to satisfy the requirement?

---

> ### Author Rebuttal · Authors · 2026-03-31
>
> We thank the reviewer for their insightful comments and concerns.
>
> **[Domain Variability Assumption]**
>
> The domain variability assumption does not require domain differences to be simple or disentangled; it only requires that style-dependent events have different probabilities across at least one pair of domains. Thus, the assumption can accommodate complex domain shifts.
> However, we may not have fully understood the second part of the question, especially the example of disentangling ID and cartoon Pixar styles. If the reviewer could clarify this point further, we would be happy to address it more precisely in the revision.
>
> **[Assumption on generator]**
>
>  The smoothness assumption on g concerns the mapping from latent factors (c, s) to the data manifold, not spatial smoothness in the image coordinates. Thus, images may contain sharp object boundaries while still varying smoothly with respect to the latent variables. For instance, object position can change smoothly in latent space even if its edge appears sharp in pixel space. More broadly, the C1 smoothness assumption is a standard idealization used for generative models and representation learning [R1- R5]. It models data generated by continuously varying latent factors on a low-dimensional manifold. If real data distribution contains singularities or discrete transitions, Proposition A.1 may not hold exactly, and the theory should then be viewed as approximate or local. We will clarify this in our revision.
>
> **[Decomposition of content and style and their overlap]**
>
>  The decomposition in L132 is a local tangent-space assumption, not a claim that every semantic feature is globally either content or style. It only requires that infinitesimal variations around a sample admit a decomposition into content-induced and style-induced directions. If a variation genuinely lies in both subspaces, then exact orthogonality is violated and identifiability is no longer guaranteed exactly. In such cases, L_orth may bias the model to assign that variation mainly to one factor or to approximate it by a nearby orthogonal decomposition. Since L_orth​ is a soft regularizer, this effect is balanced by the distribution-matching losses, which also enforce realistic image generation. Moreover, we have derived bounds and identifiability results when orthogonality is only approximately satisfied. The summarized theorem can be stated as follows:
>
> **Theorem (Approximate style identifiability).**
> Suppose all assumptions of the original theorem hold, except that exact orthogonality is replaced by the following approximate condition: the minimum principal angle between the content and style tangent subspaces is at least $\frac{\pi}{2}-\xi$, i.e.,
> $$
> \angle\big( \mathcal R(J_{c}g),~ \mathcal R(J_{s^{(n)}}g) \big)
> \ge \frac{\pi}{2}-\xi
> \qquad \forall n\in[N].
> $$
> Then
> $$\|| J_{c}\hat{s}^{(n)}\||_2 \le \frac{\sin \xi}{\sigma\_{\min} (J\_{\hat{s}} \hat{g}(\hat{c},\hat{s}^{(n)}) ) } \||J\_{c}x^{(n)}\||_2. $$
> Hence, exact style identifiability presented in paper is recovered when $\xi=0$, while for small $\xi>0$, the learned style $\hat{s}^{(n)}$ is approximately content-independent.
>
> **Proof sketch:**
> When the content and style tangent subspaces are only approximately orthogonal, i.e., their smallest principal angle is at least $\min \\{ \frac{\pi}{2} \pm \xi \\} = \frac{\pi}{2} - \xi$, the learned style may depend weakly on content, but this dependence remains controlled. We derive the bound using following steps:
> 1. First, project the Jacobian with respect to the ground-truth content $J_{c}g$ onto the learned style subspace $J_{\hat{s}}\hat{g}$. This isolates the term involving $J_{c}\delta$.
> 2. Use Eq. 22 in paper to replace the learned style subspace with the true style subspace.
> 3. Since $J_{c}g$ lies in the true content tangent space, the resulting term depends on the overlap between the true content and style subspaces.
> 4. Approximate orthogonality bounds this overlap by $\sin\xi$.
> 5. The full-column-rank property of $J_{\hat{s}}\hat{g}$ (see equation 21 in paper) then converts this into a bound on $\||J_{c}\hat{s}^{(n)}\||_2$.
>
> This gives the final approximate identifiability bound. We will present this new result in the paper.
>
> **References:**
>
> [R1] Khemakhem et. al 2020, Variational Autoencoders and Nonlinear ICA: A Unifying Framework
>
> [R2] Buccholz et. al 2022, Function Classes for Identifiable Nonlinear Independent Component Analysis
>
> [R3] Gresele et. al 2021, Independent mechanism analysis, a new concept?
>
> [R4] Hyvarinen et. al 2019, Nonlinear ICA Using Auxiliary Variables and Generalized Contrastive Learning
>
> [R5] Shen et. al 2025,  Controllable Video Generation with Provable Disentanglement

---

> > ### Author Rebuttal · Reviewer_YeXN · 2026-04-03
> >
> > Thanks for the authors to provide more details, my concerns are addressed, including the second part of my first question. I will maintain my score.

---

### Official Review · Reviewer_VE6H · 2026-03-13

**Soundness:** 3
**Presentation:** 3
**Significance:** 3
**Originality:** 3
**Overall Recommendation:** 4
**Confidence:** 3

**Summary:**

This paper studies the problem of identifying content and style from unpaired multi-domain data using generative models. The authors propose a new identifiability condition i.e., differential independence, which assumes that infinitesimal variations in content and style variables span orthogonal subspaces of the data manifold. Under this assumption and a distribution-matching objective, the paper proves that both content and style variables are identifiable up to invertible transformations.  The authors further introduce CSDI-GAN, which augments a GAN architecture with a Jacobian orthogonality regularizer approximated via Hutchinson trace estimation. Experiments on synthetic MNIST and real image datasets (AFHQ, CelebA-HQ) demonstrate improvements in counterfactual generation and domain translation tasks compared to several content-style GAN baselines.

**Compliance With Llm Reviewing Policy:**

Affirmed.

**Final Justification:**

During the rebuttal, the authors presented new theoretical results on their method and empirical results on computational overhead, which clarify my previous concerns. Their also discussed the choice of experiment scale and applicability of the method to diffusion models, which are helpful to understand the limitation and the core contribution of this work. Thus, I raise the significance score from 2 to 3 and maintain my positive score for the overall recommendation.

**Key Questions For Authors:**

- is the method sensitive to violations of the assumption on exact orthogonality of Jacobian subspaces?
- how does the computational overhead scale with model size and image resolution? compare the training costs of CSDI-GAN and baseline GAN methods would be useful.
- would the differential independence still be meaningful to score-based diffusion models, where mapping from latents to data is implicit?

**Limitations:**

yes

**Strengths And Weaknesses:**

Strengths:
- the geometric notion of disentanglement is conceptually appealing and aligns with recent work on causal representation learning and differential geometry
- the identifiability theorem suggests content and style can be recovered without assuming statistical independence, which addresses the limitation of prior work
- translating the differential independence condition into a regularizable loss function implemented in a GAN is an interesting idea.

Weaknesses:
- the orthogonallity assumptions on content and style-induced tangent spaces might still be too strong for natural generative processes.
- the evaluation setup is modest, with the datasets being relatively small scale, and no comparison with recent disentanglement approaches beyond GAN variants.
- most results rely on FID and LPIPS, which primarily measure image quality rather than disentanglement fidelity.

---

> ### Author Rebuttal · Authors · 2026-03-31
>
> We thank the reviewer for their valuable comments, please find our responses below.
>
> **[Orthogonality assumptions, robustness of our criterion on CSDI violations]**
>
> Regarding how strong the CSDI assumption is, our sense is as follows. First, similar Jacobian orthogonality assumptions are well-motivated from perspective of disentanglement, and they’re empirically justified; see StyleGAN2 path length regularization [R1], VAEs with Gaussian priors [R2], and others in Remark 3.2. Second, content-style separation used to rely on statistical independence [R3, R4] or Jacobian sparsity [R5], which can be stringent in some cases.
> We also agree that exact orthogonality as in Assumption 3.1 may not hold in practice. To address this, we derived a new theoretical result and experiment showing how violations of CSDI affect identifiability.
> Due to space limitations, we refer reviewer to
> 1) the reply to reviewer YeXN for proof sketch where we show the approximate identifiability under orthogonality violation and,
> 2) the reply to reviewer xgMt on the sensitivity analysis of orthogonality assumption.
>
> **[Evaluation setup & Datasets]**
>
> Due to compute constraints, our experiments have a modest setup. Nonetheless, we hope to stress that our focus lies in “theory validation”, and current experiments do support our core theory. Regarding baselines, we focus on GAN-based methods as our implementation is instantiated in a multi-domain GAN framework, making them the most direct comparisons for domain translation and counterfactual generation. Many recent disentanglement approaches are designed for different settings, such as single-domain data, paired supervision, or objectives that do not explicitly address unpaired multi-domain content–style identifiability. That said, we agree that a broader empirical comparison would strengthen the paper, and we will clarify this limitation explicitly in the revision.
>
> **[Metrics for Disentanglement]**
>
> We agree that FID & LPIPS mainly measure image quality and style diversity, rather than disentanglement fidelity directly. However, for real datasets such as AFHQ and CelebA-HQ, ground-truth content & style factors are unavailable, so disentanglement cannot be measured. In this setting, FID & LPIPS serve as practical proxy metrics for evaluating whether the learned representation supports high-quality domain translation & counterfactual generation.
> For our MNIST experiment, where ground-truth content $c$ and style $s^{(n)}$ are available, we additionally evaluate disentanglement directly using the $R^2$ score of the ground-truth vs learned content $\hat c$ & style $\hat s^{(n)}$. Specifically, we train a linear regressor to predict ground-truth $c$ from $\hat c$, and $s^{(n)}$ from $\hat s^{(n)}$.
>
> | Method | $R^2$ content | $R^2$ style (avg. across domains) |
> | :--- | :---: | :---: |
> | CSDI w\o orthogonality | 0.7324 | 0.5618 |
> | BI | 0.7744 | 0.7338 |
> | CSDI | 0.7675 | 0.8995 |
>
> CSDI achieves a much higher $R^2$ score for style and a competitive score for content compared with baselines. This suggests that the proposed orthogonality constraint is crucial to accurately learn the style. We will keep this table in revision.
>
> **[Computational Overhead]**
>
> We thank the reviewer for this suggestion. We refer the reviewer to Remark 4, where we analyze the memory overhead of our method with respect to model size & image resolution. We additionally report the wall-clock training time & peak GPU memory for CSDI-GAN & the GAN baselines over 720K iterations on AFHQ using a V100 GPU.
>
> | Method | Training time (hours) | Peak GPU memory (GB) |
> | :--- | :---: | :---: |
> | StyleGAN | 23.11 | 9.38 |
> | BI-GAN | 28.32 | 12.65 |
> | I-GAN | 28.90 | 11.30 |
> | CSDI-GAN | 30.09 | 14.20 |
>
> Results show that CSDI-GAN has a moderate overhead due to our Jacobian regularization. Importantly, this additional cost is accompanied by improved FID/LPIPS. We will add this result in the revision to clarify the computational tradeoff.
>
> **[Applicability of CSDI in Score-based Generative Models]**
>
> Yes, conceptually, CSDI is applicable for diffusion models. In principle, our identifiability result is agnostic to the implementation framework (e.g., GANs, VAEs, or diffusion/flow). But as score-based methods do not have an explicit generator like x = g(c, s), the implementation details may require substantial re-design to realize our identifiable formulation in Eq 3. We believe such effort would warrant a standalone work. We will mention this in our conclusion section.
>
> **References**
>
> [R1] Karras et al. 2020, Analyzing and improving the image quality of StyleGAN
>
> [R2] Rolinek et al. 2019, Variational autoencoders pursue pca directions
>
> [R3] Xie et al 2023,  Multi-domain image generation and translation with identifiability guarantees
>
> [R4] Shrestha et al. 2025, Content-style learning from unaligned domains: Identifiability under unknown latent dimensions
>
> [R5] Yan et al. 2024, Counterfactual generation with identifiability guarantees

---

> > ### Author Rebuttal · Reviewer_VE6H · 2026-04-04
> >
> > Thanks for the detailed rebuttal. I appeciate the added theoretical and empirical results. the computational overhead results and discussions on the experiment scale & applicability to diffusion models are helpful to clarify the core contribution this work. I encourage them to add them in the revision. Hence, I will maintain my positive scores.

---

### Decision · Program_Chairs · 2026-04-30

**Decision:**

Accept (regular)

**Comment:**

Summary: this work investigates the problem of identifying content and style from unpaired multi-domain data using generative models, which is an important topic. The authors propose a new identifiability condition, which assumes that infinitesimal variations in content and style variables span orthogonal subspaces of the data manifold. Under this assumption and a distribution-matching objective, the paper proves that both content and style variables are identifiable up to invertible transformations. The authors further introduce CSDI-GAN, which augments a GAN architecture with a Jacobian orthogonality regularizer approximated via Hutchinson trace estimation. Experiments on synthetic MNIST and real image datasets (AFHQ, CelebA-HQ) demonstrate improvements in counterfactual generation and domain translation tasks compared to several content-style GAN baselines. This is an interesting work that I believe is of interest to the community.